# Global evidence of positive biodiversity effects on spatial ecosystem stability in natural grasslands

Yongfan Wang[1], Marc W. Cadotte [2,3], Yuxin Chen[1], Lauchlan H. Fraser[4], Yuhua Zhang[1,5], Fengmin Huang[1], Shan Luo[1], Nayun Shi[1] & Michel Loreau[6]

The effect of biodiversity on primary productivity has been a hot topic in ecology for over 20 years. Biodiversity–productivity relationships in natural ecosystems are highly variable, although positive relationships are most common. Understanding the conditions under which different relationships emerge is still a major challenge. Here, by analyzing HerbDivNet data, a global survey of natural grasslands, we show that biodiversity stabilizes rather than increases plant productivity in natural grasslands at the global scale. Our results suggest that the effect of species richness on productivity shifts from strongly positive in low-productivity communities to strongly negative in high-productivity communities. Thus, plant richness maintains community productivity at intermediate levels. As a result, it stabilizes plant productivity against environmental heterogeneity across space. Unifying biodiversity–productivity and biodiversity–spatial stability relationships at the global scale provides a new perspective on the functioning of natural ecosystems.

[1] State Key Laboratory of Biocontrol, Guangdong Key Laboratory of Plant Resources, Department of Ecology, School of Life Sciences, Sun Yat-sen University, 510275 Guangzhou, China. [2] Department of Biological Sciences, University of Toronto-Scarborough, 1265 Military Trail, Toronto, ON M1C 1A4, Canada. [3] Department of Ecology and Evolutionary Biology, University of Toronto, 25 Willcocks St., Toronto, ON M5S 3B2, Canada. [4] Department of Natural Resource Sciences, Thompson Rivers University, Kamloops, BC V2C 0C8, Canada. [5] Biology and Food Engineering Institute, Guangdong University of Education, 510303 Guangzhou, China. [6] Centre for Biodiversity Theory and Modelling, Theoretical and Experimental Ecology Station, CNRS, 09200 Moulis, France. Correspondence and requests for materials should be addressed to Y.W. (email: lsswyf@mail.sysu.edu.cn) or to M.L. (email: michel.loreau@cnrs.fr)

The effect of plant biodiversity on primary productivity has been a central research theme in ecology over recent decades[1–6], but the variation in the strength and shape of biodiversity–productivity relationships has led to debates about the generality of these patterns and their causal mechanisms[4,7–13]. Changes in biodiversity–ecosystem functioning relationships along environmental gradients have been reported in both biodiversity experiments[14,15] and natural plant communities[5,12,16–18], although a positive relationship is the most common pattern[1–5,12,19].

A number of mechanisms have been proposed to explain why biodiversity–productivity relationships may be context-dependent[18]. At the global extent, however, stress level might be the most important factor in modulating biodiversity effects on productivity, because: (1) stress is a broad concept that includes light, water, and nutrient availability, suboptimal temperatures, and disturbances[20,21], and hence that encompasses most of the factors that affect plant growth; and (2) the effects of biodiversity on ecosystem functioning come from species interactions, the magnitude and direction of interactions of which shift along stress gradients[14,22]. A recent study showed that water availability was the most important factor modulating the relationship between species richness and forest functioning[18]. Although stress is hard to measure precisely[23,24], stressful environments might be best defined as those in which plants are limited by the environment in their ability to convert energy into biomass[20,24,25]. Therefore, we can assume that high stress is associated with low productivity. Changes in the sign and shape of the biodiversity–productivity relationship along gradients of stress or productivity have rarely been investigated[16,26].

The stress-gradient hypothesis predicts that facilitative interactions amongst plant species dominate under harsh conditions, whereas competitive interactions dominate under more favorable conditions[16,21,22,27,28]. The complementarity effect includes niche partitioning and facilitative interactions between species[29]. For example, complementarity between species is more important in the more stressful environment of boreal forests than in temperate forests growing in a more stable and productive environment where competition is more prevalent[16]. Community productivity mainly depends on the yield of dominant species (competitively superior species), but dominant species are not necessarily high-yielding species[30]. As a result, the role of the selection effect in the relationship between biodiversity and productivity is generally variable, while the complementarity effect is the main source of the positive effect of biodiversity on community productivity[3,31]. A decrease in the complementarity effect should weaken the positive correlation between biodiversity and productivity; thus, we may expect the positive biodiversity–productivity relationship to vary along stress or productivity gradients in response to changes in species interactions[10,16,26]. Complementarity in stressful environments might enhance the positive effect of biodiversity on productivity where average productivity is low. This positive biodiversity–productivity relationship should gradually flatten out in more fertile environments, in which complementarity is less important[16]. If this is the case, biodiversity–productivity relationships should converge as plant biodiversity increases (Fig. 1a), and thus the spatial variation in primary productivity between different communities should be smaller at high vs. low biodiversity. Therefore, our hypothesis is that biodiversity reduces the spatial variability —i.e., increases the spatial stability or predictability— in productivity at large spatial scales (Fig. 1b).

Stability has many different definitions in ecology[32,33]. Here we focus on spatial stability, $S$, which measures the similarity of ecosystem properties such as productivity across different grassland communities worldwide. $S$ is defined as 1/variability, where variability is a measure of the magnitude of spatial changes in an

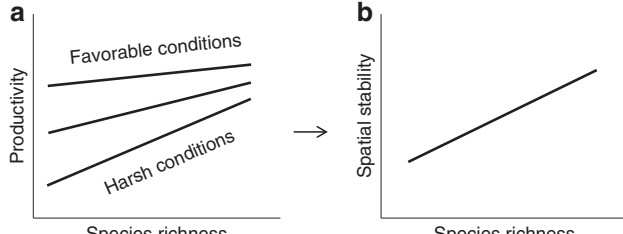

**Fig. 1** Hypothetical relationships between richness, productivity and stability in natural ecosystems. **a** The local effect of species richness on primary productivity is expected to vary along environmental gradients, leading to a convergence of richness–productivity relationships with increasing diversity. **b** As a result, the spatial stability of productivity between communities that experience different environments should increase as plant richness increases

ecosystem property. Compared with the large number of studies on the influence of biodiversity on the temporal stability of ecosystems, i.e., the invariability of an ecosystem property over time[34–40], the potential impact of biodiversity on the spatial stability of ecosystem properties has received little attention[39,41,42].

Here, we use data from a global survey of natural grasslands, the Herbaceous Diversity Network (HerbDivNet)[43,44], to test the following two hypotheses: (i) as productivity increases, the positive effect of biodiversity on plant productivity decreases (see Fig. 1a); and (ii) the spatial stability of community productivity increases with increasing biodiversity (see Fig. 1b). HerbDivNet involves five grassland types: temperate, temperate wet meadow, Mediterranean, tropical and subtropical, and alpine. The network has 9640 plots (each 1m × 1m) distributed across 151 grids (each 8m × 8m) in 30 natural grasslands (sites) from 19 countries and six continents (Fig. 3 and Supplementary Fig. 1). In each plot, both species richness and aboveground live biomass at peak biomass were measured. We use peak biomass as a surrogate measure of annual primary productivity and use plant species richness as the measure of biodiversity. Each grid was established in a patch of relatively homogeneous area, which means that each grid was small enough to exclude significant variations in abiotic conditions, which allowed us to test the effects of biodiversity on productivity. We show that as productivity increases, the positive correlation between biodiversity and productivity decreases gradually, and finally becomes negative. Our analysis further indicates that at the global extent, the role of biodiversity is not so much to promote productivity as to stabilize it against environmental heterogeneity across space.

## Results

**Biodiversity and productivity**. We first used structural equation models to test the linkages among biodiversity, productivity, and abiotic conditions. We found that the three climatic variables (total amount of precipitation, average number of daylight hours, and average temperature during the growing season) had significant effects on both biodiversity and productivity, and there was a partial negative effect of biodiversity on productivity (Fig. 2). Only 10% of the variance in productivity could be explained by number of daylight hours, temperature, precipitation, and biodiversity, and only 5% of the variance in biodiversity could be explained by the three climatic variables, suggesting that soil factors may also have important effects on biodiversity and productivity.

We then divided the 151 grids into three classes of mean productivity (low, medium, and high) with equal numbers of grids to form a productivity gradient, and repeated the structural

equation model analysis for the three groups separately (Fig. 3). In this way, we were able to test how the partial effect ($r_\partial$) of biodiversity on productivity changed across the productivity gradient. We found that $r_\partial$ differed more considerably among productivity classes than our hypothesis predicts (see Fig. 1a). There was a clear tendency for $r_\partial$ (mean ± SD) to shift from strongly positive ($0.247 \pm 0.019$) to weakly positive ($0.089 \pm 0.019$) to strongly negative ($-0.307 \pm 0.017$) under unproductive, moderate and highly productive conditions, respectively (inset in Fig. 3). When we consider all three productivity groups together by using a multi-group modeling, the results were similar: $r_\partial = 0.247 \pm 0.005$, $r_\partial = 0.089 \pm 0.006$, and $r_\partial = -0.307 \pm 0.022$ for low, medium, and high productivity, respectively. The effects of biodiversity on productivity also showed large variations among the 151 grids and three grid groups (grids were divided into low, medium, and high productivity groups as above) in the Bayesian models. Grids (Figs. 4a and 5) or grid groups (inset in Fig. 5) with a higher productivity tended to have more negative biodiversity effects, while grids or grid groups with a lower productivity tended to have more positive biodiversity effects.

**Biodiversity and spatial variability**. The opposite biodiversity–productivity relationships at low and high productivity suggest that biodiversity does not increase plant productivity universally but instead stabilizes it across space. Therefore, we further investigated the global relationships between biodiversity and the spatial variability of plant productivity. Plots were grouped by species richness, and SD (standard deviation) and CV (coefficient of variation) of productivity across plots were then calculated within each richness group. Using simple linear and quadratic regressions, we found that increasing biodiversity significantly reduced productivity variability (Fig. 6a, c). Also, the negative relationship between biodiversity and productivity variability was not sensitive to the statistical method used: we constructed hierarchical Bayesian models to fit both the central tendency of productivity (i.e., biodiversity effect on mean productivity) and the variation around the central tendency (i.e., biodiversity effect on productivity variability) as a function of biodiversity, and SD and CV of productivity all decreased with biodiversity (Figs. 4b and 6b, d and Supplementary Table 1).

We then tested if the biodiversity–spatial variability relationship depended on the level of productivity. The 151 grids were divided into three equal groups (low, medium, and high) depending on their mean productivity as in Fig. 3. We repeated the simple regression analysis (Supplementary Fig. 2) and the hierarchical Bayesian model (Supplementary Fig. 3) for the three groups separately. The biodiversity–spatial variability relationship shifted from slightly negative (or no correlation) at the low

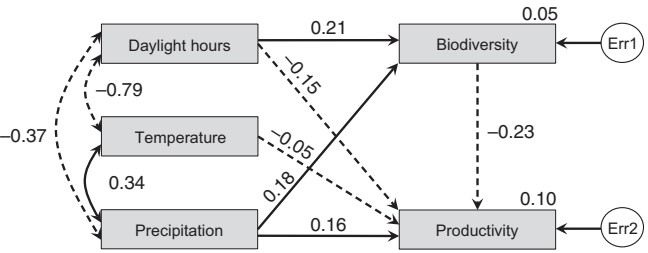

**Fig. 2** Structural equation models showing the connections between biodiversity and productivity at the plot level ($N = 9640$ plots). All retained arrows are significant ($P < 0.05$). Solid and dashed one-way arrows represent positive and negative effects, respectively. Solid and dashed two-way arrows represent positive and negative correlations, respectively. Standardized regression weights (along one-way arrows), correlations (along two-way arrows) and squared multiple correlations (beside Biodiversity and Productivity boxes) for the fitting model are shown. The exogenous unobserved variable err1 and err2 account for the unexplained errors in the estimation of biodiversity and productivity, respectively. Test statistic = 0.761, with 1 model degree of freedom and $P = 0.383$ (indicating close model-data fit). Source data are provided as a Source Data file

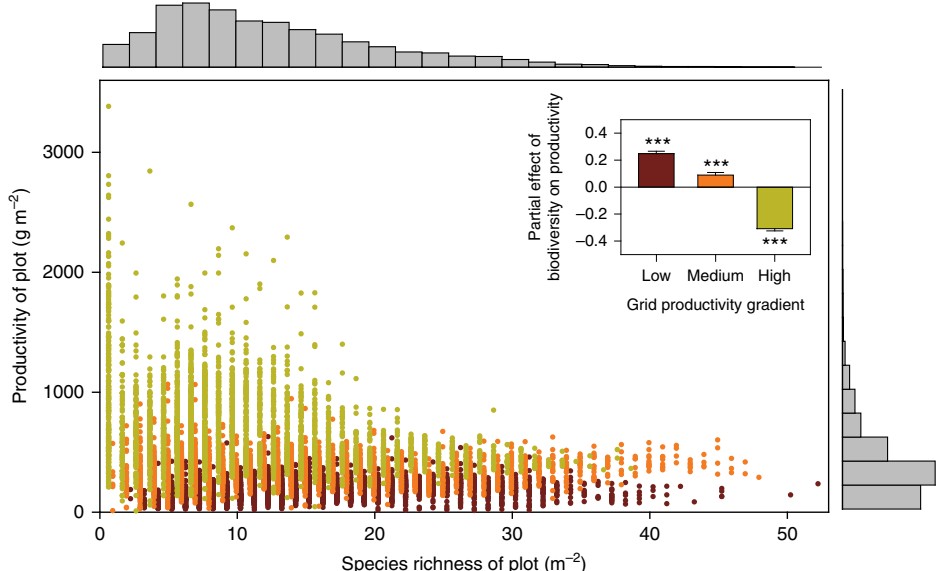

**Fig. 3** Global relationships between biodiversity and productivity in natural grasslands at the plot level. Marginal histograms show the frequency distribution of biodiversity and productivity across plots ($N = 9640$ plots). The 151 grids were divided into three equal groups depending on their mean productivity: low, medium, and high productivity (with 50–51 grids each), corresponding to different colors of the points (plots). Repeating the structural equation model analysis for the three groups separately, the partial effects (mean ± SD) of biodiversity on productivity along the productivity gradient are shown in the inset. Each bar corresponds to points of the same color. Symbols are slightly displaced at x-axis to improve readability. Asterisks indicate a significant difference from zero: *$P < 0.05$; **$P < 0.01$; ***$P < 0.001$. Source data are provided as a Source Data file

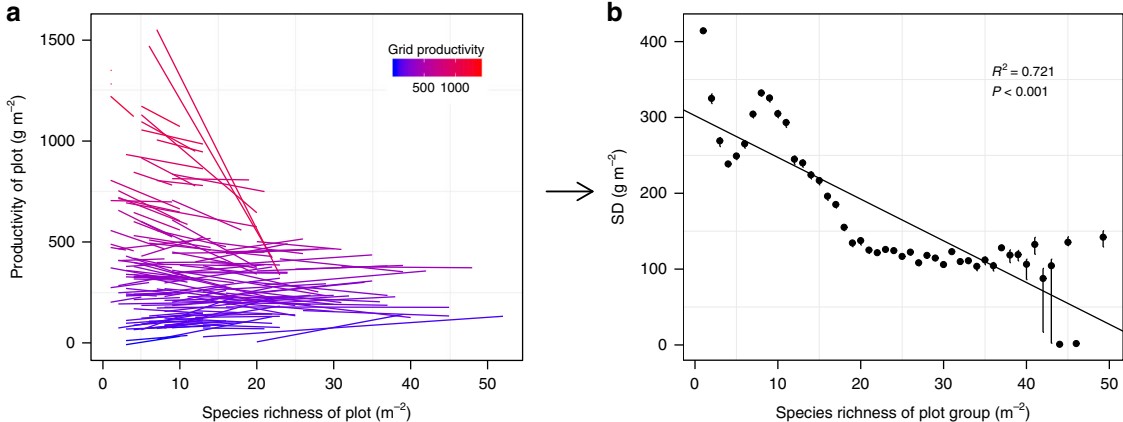

**Fig. 4** Relationships between species richness, productivity and spatial variability of productivity. **a** Relationships between richness and productivity varied among 151 grids with different average productivity or stress gradients in natural grasslands. **b** Standard deviation (SD) of productivity (i.e., inverse of spatial stability) decreased with species richness in global natural grasslands. Black points and vertical lines show the medians and 95% credible intervals (CI) of productivity variation for each species richness level. Source data are provided as a Source Data file

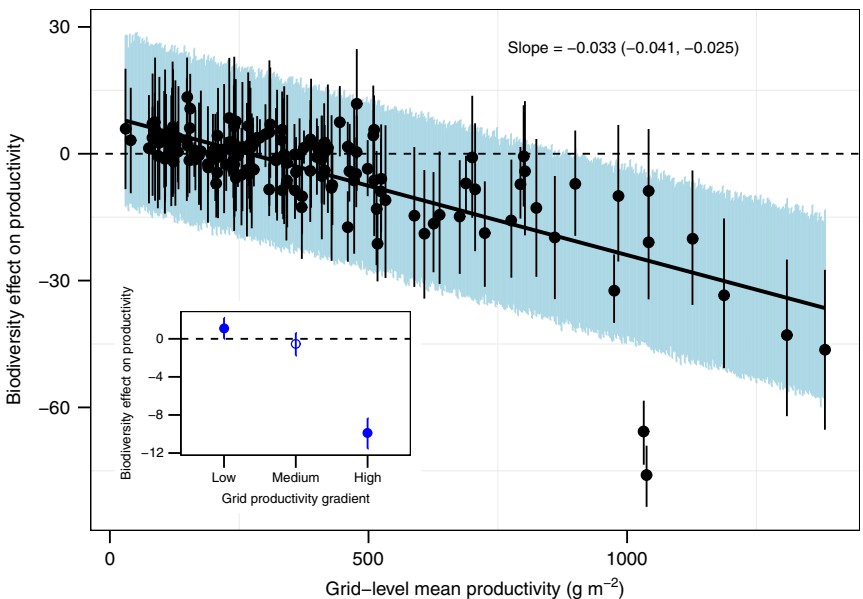

**Fig. 5** Shift in the sign and strength of the biodiversity effects across productivity gradients in hierarchical Bayesian models. Each point represents the mean productivity (horizontal axis) and the median value of biodiversity effect on productivity (vertical axis) at each grid ($N = 151$ grids) or grid group (low, medium, and high productivity groups in the inset). Vertical lines represent the 95% of credible intervals (CI) of estimated biodiversity effects. The solid line is the estimated relationship between mean productivity and biodiversity effect. The shaded area is the 95% CI of the estimated relationship. The text shows the median and 95% CI of the slope. The filled blue circles in the inset represent significant results. Source data are provided as a Source Data file

productivity level to increasingly negative at the medium and high productivity levels. In other words, the higher the productivity level, the stronger stabilizing effects of biodiversity across space. Although the trends were similar overall, the results were slightly different depending on whether variability was measured by SD or CV. The relationship between biodiversity and spatial variability was fairly negative in the 'high productivity' group, weakly negative in the 'medium productivity' group, and non-significant for the 'low productivity' group when using SD. When using CV, all the relationships were significantly negative.

Structural equation models revealed that species richness had a strong negative effect on productivity spatial variability ($r_{\partial} = -0.39$, $P \leq 0.05$ for both SD and CV). More than 78% of the variance in productivity spatial variability was explained by the number of plots, species richness and mean productivity (Fig. 7, Supplementary Fig. 4 and Supplementary Table 2; see

Methods). Similar results were obtained when we used grid-mean data (Supplementary Fig. 5). We also tested if more diverse grids were less variable in productivity across plots. Both SD and CV of productivity across plots for each grid were negatively correlated with grid-level mean richness (Supplementary Fig. 6). Thus, our results show unambiguously that grassland productivity becomes less variable or more predictable when species richness increases, regardless of abiotic conditions.

## Discussion
At the global extent, by analyzing HerbDivNet data across 30 sites in 19 countries and six continents, our results reveal that the shape of the biodiversity–productivity relationship depends strongly on the level of productivity of natural grasslands. Under low-productivity conditions (corresponding to stressful habitats),

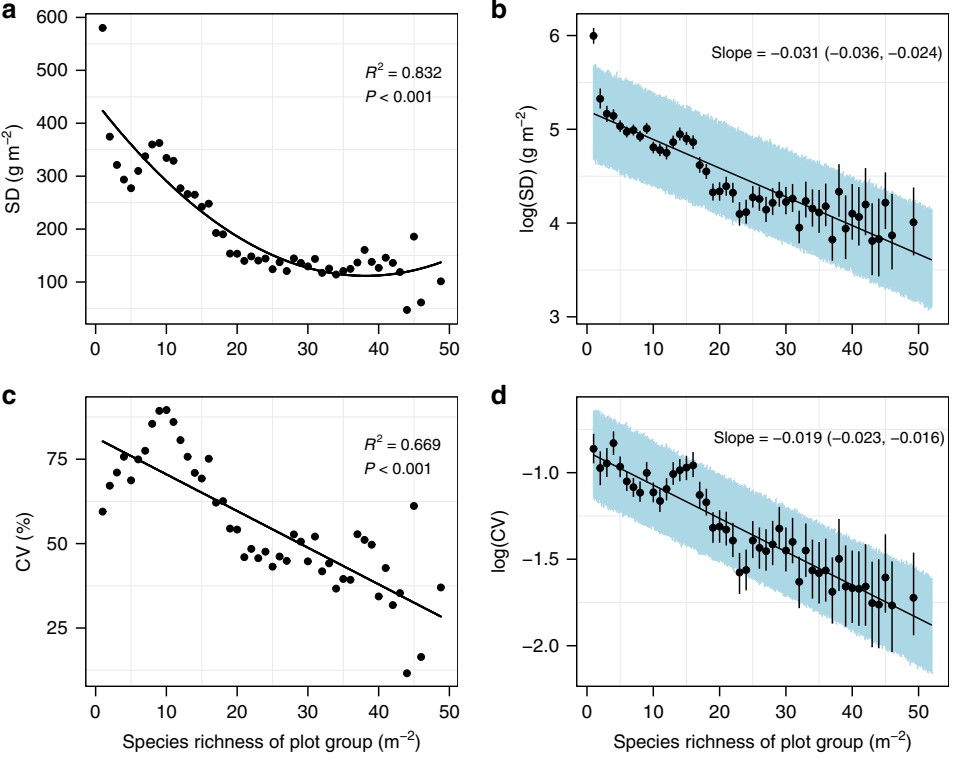

**Fig. 6** Global relationship between biodiversity and spatial variability ($N = 47$ plot groups). **a**, **c** Simple regression models. **b**, **d** Hierarchical Bayesian models. Black points and vertical lines in **b**, **d** show the medians and 95% credible intervals (CI) of log-transformed productivity variation for each species richness level. Spatial variability in grassland productivity was measured as standard deviation (SD, top panel) or coefficient of variation (CV, bottom panel) across plots with the same richness. The black line represents the fitted relationship. The shaded areas in panels **b** and **d** represent the 95% CI of the fitted relationships. The texts in panels **b** and **d** show the median and 95% CI of the slope. Species richness was used to group plots containing one, two, three species, etc. Because some high-richness levels had only one or two plots at each richness level, the 9640 plots were divided into 47 plot groups, with the last group containing 47−52 species, to ensure there were three or more plots in each group. Source data are provided as a Source Data file

biodiversity promotes community productivity; while under high-productivity conditions (corresponding to favorable environments), biodiversity inhibits community productivity. Our results may be due to variations in the sign and strength of interspecific interactions along the stress gradient. Because community productivity was not separated by species and species productivity in monocultures was not available[44], we were unable to directly analyze variations in species interactions along productivity gradients. More detailed analyses of interspecific interactions would be useful in future research.

Small-scale experiments using artificial grasslands have repeatedly found positive effects of biodiversity on productivity[1,2,13,45]. We suggest that artificial grasslands rarely include low- and high-productivity ecosystems. Ecologists often look for 'suitable' sites for their experiments. The environment cannot be too fertile (leading to overgrowth) or too harsh (making it difficult to establish grassland vegetation in a short time). Thus, the range of environmental conditions in experimental grasslands may be much narrower than the range in natural grasslands. For example, BIODEPTH is one of the most famous artificial grassland experiments conducted in eight sites in seven European countries[1]. Like 'grid' in HerbDivNet, each 'site' in BIODEPTH was also established in a patch of relatively homogeneous area. Using the mean peak biomass of plots with eight species (the highest richness common to all eight sites), as an approximation of the mean productivity of the site, the range of productivity in BIODEPTH (site-level mean aboveground biomass ranged from 337 to 802 g m$^{-2}$) was much narrower than that of HerbDivNet (grid-level mean aboveground biomass ranged from 30 to 1382 g m$^{-2}$). Therefore, a wide range of productivity (a sufficiently large environmental stress gradient)

may be necessary to see the full picture of changes in biodiversity–productivity relationships. This might be a reason why grassland experiments rarely detected negative biodiversity effects on productivity. Indeed, our high-productivity grids (12 grids with mean biomass over 850 g m$^{-2}$, ~8% of the 151 grids) contributed considerably to the negative effects of biodiversity on productivity (Fig. 5).

Although biodiversity did not have a consistent effect on productivity, it did contribute to the stability of productivity across space through changes in the biodiversity–productivity relationship along a productivity gradient. Our work suggests that biodiversity stabilizes rather than increases productivity at large spatial scales in natural grasslands. Biodiversity appears to buffer ecosystems against environmental heterogeneity, and thus to reduce their dependency on abiotic factors[46]. Furthermore, the species richness–spatial variability relationship varied with the productivity level, i.e., species diversity increased spatial stability, and these effects were greatest in the most productive ecosystems. This is complementary to the results of the world's longest-running experiment across ecosystems[47], which considered temporal stability. The relationships between richness and spatial variability had consistently negative slopes along the productivity gradients when using CV as a measure of variability, but this was not the case when using SD (see Supplementary Figs. 2 and 3). This is due to the difference in the measurement of spatial variability: SD is a measure of the average deviation from the mean productivity, and is thus an absolute measure of variability; while CV is the ratio between SD and mean productivity, and is thus a relative measure that removes the impact of mean productivity. In the high productivity group, the relationship

**a**

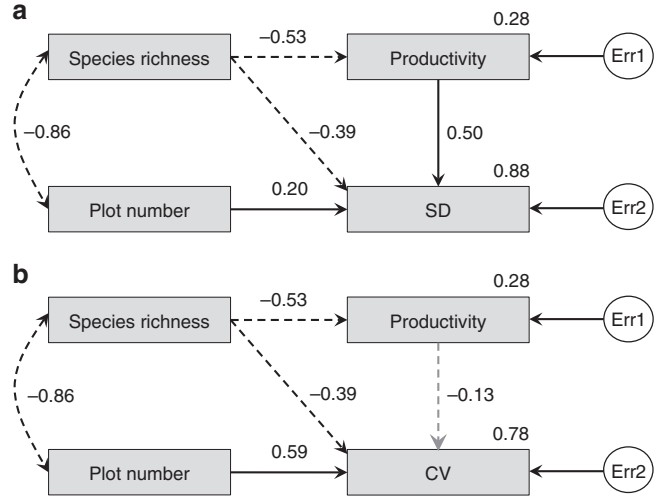

**b**

**Fig. 7** Structural equation model showing the connections between biodiversity, productivity, and variability. The grouping of plots is consistent with Fig. 6 (N = 47 plot groups). Variability was measured either as the standard deviation (**a**), or as the coefficient of variation (**b**) of productivity across plots within each level of species richness. Solid and dashed one-way straight black arrows represent positive and negative effects, respectively. The one-way gray arrow represents an insignificant ($P > 0.05$) effect. Dashed two-way black arrows represent negative correlations. To avoid zero degree of freedom, the 'productivity' variable in **b** was not dropped from the model. The exogenous unobserved variables err1 and err2 accounted for the unexplained errors in the estimation of productivity and productivity variability, respectively. Productivity data was ln-transformed before analyses to ensure normality. Test statistic = 0.863, with one model degree of freedom and $P = 0.353$ for both, indicating close model-data fit. Standardized regression weights (along one-way arrows), correlations (along two-way arrows) and squared multiple correlations (beside Productivity, SD, and CV boxes) for the best-fitting model are shown. Source data are provided as a Source Data file

between richness and productivity was negative (see the insets in Figs. 3 and 5), thus reducing the negative correlation between richness and spatial variability when the measure of variability is changed from SD to CV. There is still a fairly negative correlation between richness and CV, indicating that our result that biodiversity increases spatial stability is very robust in highly productive environments. However, as the level of productivity decreases (in medium and low productivity groups), the relationship between richness and productivity becomes positive, which contributes to strengthening the negative correlation between richness and variability when SD is replaced by CV. Since CV removes the impact of mean productivity, it is a more widely used measure of ecosystem variability[34,35,39,41,42], which suggests that our conclusion that biodiversity decreases spatial variability is robust. These findings have important implications in the context of global change. As a result of global warming and rising carbon dioxide concentrations in the atmosphere, the primary productivity of ecosystems is expected to increase as well. Larger numbers of species will be needed to maintain ecosystem performance as the global climate continues to change[48,49].

With the decrease of environmental stress (that is, as productivity increases), the positive correlation between biodiversity and productivity weakens gradually, and eventually becomes negative. This gradual change explains why spatial stability increases with increasing biodiversity. Existing theories, however, do not fully explain the negative relationship between biodiversity and productivity. Two primary mechanisms, the complementarity and selection effects, have been widely used to

interpret biodiversity effects on ecosystem functioning. The complementarity effect results from resource partitioning or positive interactions between species[29], and is generally positive[3,29,31]. The selection effect is due to shifts in dominance driven by interspecific competition[29,50]. The net biodiversity effect (complementarity+selection) on productivity may be negative if low-yielding species tend to be dominant in plant communities, which leads to a negative relationship between biodiversity and productivity. Experimental researches have shown that a priority effect due to species growing at different times explains this negative correlation between success in interspecific mixtures and biomass production. Low-yielding grass species grow early in the year and suppress higher yielding forbs that grow later[30,51], it leads to negative selection and complementary effects[30]. This could continue year after year, if grasses have earlier phenology than forbs, and they suppressed growth of the forbs[51]. It is true that high-yielding species are not always the most dominant[30,51]. This mechanism partially explains the negative relationship between biodiversity and productivity in all environments.

An alternative hypothesis that could explain the negative effect of biodiversity at high levels of productivity is allelopathic effects at high plant density. To compete for space and resources, many plant species interfere with neighboring plants through the production and release of secondary metabolites (allelochemicals) in the surrounding environment through leachates, litter decomposition, root exudates and leaf volatilization[52]. Communities with a higher productivity may also experience stronger interspecific competitive interactions, which may increase allelopathic effects. Mutual poisoning among species will reduce the productivity of the plant community as a whole. If allelopathy has larger negative effects on plant growth at high vs. low biodiversity under fertile conditions, it could create a negative relationship between biodiversity and productivity. We hope that our work will be a starting point for new experiments to test this hypothesis. Understanding the biological mechanisms that underlie negative biodiversity effects on productivity will be an important addition to the existing theory of biodiversity and ecosystem functioning.

A few experimental studies have also shown that biodiversity decreases spatial variation in ecosystem properties[41,53,54], but these studies were all based on artificial or highly manipulated ecosystems at very small scales and over limited richness gradients. Varying relationships between biodiversity and ecosystem functioning along environmental stress gradients have also been found in other natural systems. By reanalyzing data from a global survey of drylands[55], Jucker & Coomes[26] showed that the strength of the relationship between biodiversity and ecosystem multifunctionality changed consistently from slightly negative to increasingly positive as the environment became harsher. The scope of this survey was limited to arid, semi-arid, and dry-subhumid ecosystems, which collectively cover 41% of Earth's land surface. Another large-scale forest survey in eastern Canada also revealed that different biodiversity–productivity relationships in boreal and temperate forest types converged as plant species diversity increased once climate and environment were factored in Paquette et al.[16]. Temperate forests with favorable habitats showed a nearly flat relationship between biodiversity and productivity but a proportionally larger intercept (i.e., a greater productivity at low diversity), while boreal forests in less favorable environments showed a lower productivity on average but a much stronger, mostly positive linear response to biodiversity. Although these two studies[16,26] made no further analysis to link the convergence of the different biodiversity–functioning relationships to spatial stability, the trend is clear. Both results from natural drylands and forest ecosystems support our hypothesis

that, on a large scale, the effect of biodiversity is not so much to enhance ecosystem functioning as to stabilize it.

We acknowledge that in addition to light, temperature, and water, other abiotic and biotic factors also have important effects on plant richness and productivity. For example, Grace et al.[4] showed that soil factors, and the influences of human and herbivorous animals are important determinants of grassland richness, biomass, and productivity. We did not have access to data for these variables, and thus we did not include them in our analyses.

There is some evidence that biodiversity increases the stability of ecosystem processes and properties through time, across space, or both[34–41,47,53,54]. Although there has been some overlap between studies on biodiversity–productivity and biodiversity–stability relationships, the connections between the effects of biodiversity on ecosystem functioning and ecosystem stability remain unclear[3,10,35,38,56,57]. Our work provides a unifying perspective on these two types of relationships. The effects of biodiversity on productivity and on its spatial stability should be viewed as two aspects of the same ecological process, i.e., the variations in the sign and strength of interspecific interactions along a stress or productivity gradient. Our findings have important implications for biodiversity conservation. They suggest that biodiversity conservation will be more beneficial to mean ecosystem functioning in low-productivity areas, while it will be more beneficial to the stabilization of ecosystem functioning in high-productivity areas. They also suggest that the effects of biodiversity on ecosystem functioning can only be understood comprehensively on large, or even global, scales. Integrating biodiversity–productivity and biodiversity–stability relationships into a single unified picture at the global extent provides a system-level understanding of ecological processes that is likely to transform ecology into a more systematic science.

## Methods

**Data selection**. To minimize the methodological differences that exist among disparate studies, a global survey of natural grasslands uses consistent data-collection methods for testing global hypotheses in ecology and environmental science: the Herbaceous Diversity Network (HerbDivNet)[43,44]. HerbDivNet conducts coordinated surveys in 30 natural grasslands (sites) from 19 countries and six continents (Supplementary Fig. 1; the map is displayed using package "maps" in R version 3. 4. 1). Grassland type was separated into 5 categories: temperate, temperate wet meadow, Mediterranean, tropical and subtropical, and alpine. At each site, between 2 and 14 grids (each 8 m × 8 m in size) were sampled, and each grid was located in a patch of relatively homogeneous environment. HerbDivNet has 10,048 plots (each 1 m × 1 m), most of which have peak live biomass, total (live+litter) biomass and species richness data (https://datadryad.org/resource/doi:10.5061/dryad.038q8). The decomposition rate of plant litter varies greatly among different sites, so that using total (live+litter) biomass would cause a large error. Therefore, we used data from the 9640 plots in which both species richness and aboveground live biomass were measured in our analysis (Fig. 3). The aboveground live biomass at peak biomass was used as a surrogate measure of primary productivity, and the plant species richness was used as a measure of biodiversity. These plots were distributed in 151 grids, each of which contained from 60 to 64 plots. HerbDivNet spans a wide range of plant biomass (from 2 to 3374 g m$^{-2}$, with 480 plots over 1000 g m$^{-2}$, about 5% of the 9640 plots), and provides sufficient replication of plots along the productivity gradient.

**Spatial variability**. We investigated the global relationships between species richness and the spatial variability of productivity at different levels of species richness. The spatial variability of productivity within a group of plots with same richness was calculated using two metrics of dispersion: (i) standard deviation (SD), the average deviation from the mean value, which has been used as a measure of ecosystem variability in previous studies[53,54]; and (ii) coefficient of variation (CV), the ratio between the standard deviation SD and the mean $\mu$ (SD/$\mu$). CV reflects the average change of productivity per unit weight, which increases comparability among different ecosystems; it is widely used as a measure of ecosystem variability[34,35,39,41,42]. We used the two metrics to make the results more robust. Both these metrics measure the inverse of the spatial stability or predictability of total productivity.

**Structural equation model**. We used structural equation models (SEM) to test the causal linkages between biodiversity, productivity, and abiotic factors. SEM is an analytical method designed to evaluate the assumptions about complex causal networks of cause−effect relationships[58,59]. SEM has been considered in recent years to be an effective way of integrative understanding of the causal mechanisms controlling biodiversity–productivity relationships in natural systems[4]. For plant growth, light, temperature, and moisture are essential. We extracted the average monthly data on air temperature and precipitation in each site from WorldClim (http://worldclim.org/version2)[60] by mapping the site location (longitude and latitude) to the nearest 1 km × 1 km grid. Note that the 'grid' here refers to the earth surface grid, it is different from 'grid', one of the grassland sampling units. We defined 'temperature' as the average air temperature of the growing season (in the northern hemisphere from January to August; in the southern hemisphere from July to February), and 'moisture' as the total amount of precipitation in the growing season. In open grasslands, light time may be a more important factor of plant growth than is light intensity, so we chose the number of daylight hours during the growing season as a measure of light availability. We obtained the average monthly number of daylight hours at each longitude and latitude integer point from 'NASA Surface Meteorology and Solar Energy: Global Data Sets' (https://eosweb.larc.nasa.gov/cgi-bin/sse/global.cgi). Light availability at each sampling site was obtained as the weighted average of the numbers of daylight hours at the four integer longitude and latitude points around it, with a weight inversely proportional to distance. We used daylight hours, temperature, and precipitation during the growing season as independent variables to explain plot richness and plot productivity. Plot richness was an intermediate variable and plot productivity a response variable (Fig. 2). Maximum likelihood was used to estimate model parameters. In order to test how the partial effect of biodiversity on productivity changes across productivity gradient, the 151 grids were divided into three equal groups depending on their mean productivity: low, medium, and high productivity (with 50 to 51 grids each). We then repeated the above SEM analysis for the three groups separately. We also used a multi-group modeling (Simultaneous analysis of several groups) by considering the three groups at the same time. We checked whether the results were similar if we considered the three groups independently or together.

We also used structural equation models to test the causal relationships between species richness, productivity, and spatial variability of productivity. The 9640 plots were grouped into 47 groups by species richness level as in Fig. 6. For each plot group, four variables were calculated: species richness, mean productivity, productivity variability across plots (measured as either SD or CV), and number of plots. Number of plots was chosen as an independent variable because productivity variability across plots may be associated with the number of plots at a given species richness: the larger the number of plots, the greater the likelihood of extreme productivity values, and the greater the variability in plot productivity. Productivity variability was always selected as a response variable with no effect on other variables. Three competing models (Supplementary Fig. 4) were set up: richness and mean productivity were both independent variables (Model a), and either richness or productivity was an independent variable and the other an intermediate variable (Model b and Model c, respectively). Maximum likelihood was used to estimate model parameters. The three competing models were then compared by using four absolute fit indices ($\chi^2$, GFI, AGFI, RMSEA), five relative fit indices (NFI, RFI, IFI, TLI, CFI), and three parsimonious fit indices (NC, AIC, CAIC; see Supplementary Table 2 for details). Model b was the best-fitting model ($P = 0.353$; Fig. 7 and Supplementary Table 2).

**Bayesian model**. As an alternative analysis, we constructed hierarchical Bayesian models to fit both the central tendency of productivity and the variation around the central tendency as a function of species richness. The Bayesian model is complementary to the SEM, but has the advantages in avoiding categorization of the plots or grids into groups artificially. The full model is:

$$\text{Productivity}_{i,j,k} \sim \text{Normal}(\alpha_{0,j} + \alpha_{1,j}^* \text{Richness}_{i,j,k} + \alpha_{\text{site}},\ \sigma_k) \quad (1)$$

$$\log(\sigma_k) \sim \text{Normal}(\beta_0 + \beta_1 * \text{Richness}_k,\ \sigma_\beta) \quad (2)$$

$$\alpha_{0,j} \sim \text{Normal}(\gamma_{0,0} + \gamma_{0,1} * L_j + \gamma_{0,2} * P_j + \gamma_{0,3} * T_j,\ \sigma_{\alpha_0}) \quad (3)$$

$$\alpha_{1,j} \sim \text{Normal}(\gamma_{1,0} + \gamma_{1,1} * \text{Grid\_productivity}_j,\ \sigma_{\alpha_1}) \quad (4)$$

where Productivity$_{i,j,k}$ and Richness$_{i,j,k}$ are productivity and species richness of plot $i$ in grid $j$ and richness level $k$ (i.e., plots with the same richness value $k$ were at the richness level $k$), respectively. $\alpha_{\text{site}}$ is a site-level random effect, which was modeled as a normal distribution with zero mean. Grid-level productivity was modeled as a function of daylight hours ($L_j$), precipitation ($P_j$) and temperature ($T_j$). $\sigma_k$, $\sigma_\beta$, $\sigma_{\alpha_0}$, and $\sigma_{\alpha_1}$ are the standard deviations of independent normal distributions; specifically, $\sigma_k$ represents the standard deviation of the productivity at each richness level (Fig. 6b). $\beta_1$ represents the effect of richness on productivity variation (the second hypothesis in Introduction or Fig. 1b). Grid\_productivity$_j$ is the average productivity at grid $j$. $\alpha_{0,j}$ and $\alpha_{1,j}$ were used to control spatial variation in average productivity and the effects of richness on productivity possibly induced by grid-specific variation in productivity, respectively; $\gamma_{1,1}$ assessed how the richness effects

on productivity vary across stress or productivity gradients (the first hypothesis in Introduction or Fig. 1a) (Fig. 5).

We assessed the goodness of model fit by performing posterior predictive checks[61]. We calculated Bayesian $p$-values ($P_B$) to assess the deviance between the posterior distribution of predicted productivity ($y^{new}$) from the fitted Bayesian models and the distribution of observed productivity ($y^{observed}$). $P_B$ is defined as the probability that the predictive distribution is more extreme than the observed distribution. We used two statistics ($T(y)$: mean and CV of productivity across all plots) to summarize the distributions.

$$P_B = \text{Probability}(T(y^{new}) \geq T(y^{observed})) \qquad (5)$$

A $P_B$ close to 0.5 indicates a good model fit. We also calculated $P_B$ from the difference between predicted and observed productivity at each plot $(\text{Probability}(y^{new} \geq y^{observed}))$. All the three Bayesian $p$-values (0.508, 0.531, and 0.490 for $p$-values based on mean, standard deviation, and plot-level productivity, respectively) indicate a good fit of the Bayesian model to the observed productivity. We further assessed whether the difference of richness–productivity relationships across grids (Fig. 1a) could lead to a positive relationship between richness and the spatial stability of productivity (Fig. 1b). We first predicted the relationships between richness and productivity at each grid from the estimated grid-level parameters in the Bayesian models ($\alpha_{0,j}$ and $\alpha_{1,j}$; Fig. 4a). We then sampled 60 plots from each grid (plot numbers of grids range from 60 to 64) and calculated the SD of predicted productivity at each richness level. We repeated the sampling for 1000 times. Finally, we assessed the relationship between richness and predicted SD of productivity using simple linear regression (Fig. 4b).

We also fitted the relationship between species richness and the coefficient of variation (CV) of productivity at each richness level using a similar hierarchical Bayesian model as in Eqs 1–4 (Fig. 6d), where we replaced log ($\sigma_k$) with log (CV$_k$) in Eq. 1. We fitted all the Bayesian models using Markov Chain Monte Carlo (MCMC) sampling techniques in *JAGS 4.2.0* using the *rjags* package[62]. We set diffuse prior distributions for all parameters (see Supplementary Materials for JAGS code). We ran three parallel chains and used Gelman and Rubin's convergence diagnostics to assess parameter convergence (with a cutoff value of 1.1)[63].

**Reporting summary**. Further information on research design is available in the Nature Research Reporting Summary linked to this article.

## Data availability

The grassland data are from Fraser et al.[44]; these data have been deposited by Fraser et al. in the Dryad repository (https://datadryad.org/resource/doi:10.5061/dryad.038q8; Title: raw plot data from globally distributed sites). Temperature and precipitation data are from WorldClim (http://worldclim.org/version2), and daylight hours data are from 'NASA Surface Meteorology and Solar Energy: Global Data Sets' (https://eosweb.larc.nasa.gov/cgi-bin/sse/global.cgi). The source data underlying Figs. 2, 3, 4a, b, 5, 6a–d, 7a, b, and Supplementary Figs. 1, 2a–f, 3a–f, 5a, b, and 6a, b are provided as a Source Data file.

## Code availability

The code for the hierarchical Bayesian models and structural equation models (SEM) are available in Supplementary Notes 1 and 2, respectively.

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

## Acknowledgements

This work was generated using data from the Herbaceous Diversity Network (https://datadryad.org/resource/doi:10.5061/dryad.038q8). We are grateful to all of the participants of HerbDivNet who have contributed data. We thank Peter B. Adler, Fangliang He, Wensheng Shu, and Chengjin Chu for helpful comments during the preparation of the manuscript. Y.W. was supported by the National Natural Science Foundation of China (Project 31170398, 30970472, and 31230013). M.L. was supported by the TULIP Laboratory of Excellence (ANR-10-LABX-41) and by the BIOSTASES Advanced Grant, funded by the European Research Council under the European Union's Horizon 2020 research and innovation programme (666971).

## Author contributions

Y.W. conceived the study. Y.W. and M.L. designed the study. Y.W. wrote the first draft of the manuscript with inputs from Y.C. Y.W., Y.C., and N.S. analyzed the data. M.L., M.W. C., and L.H.F. helped frame the study and contributed to the writing. All authors provided substantial comments on an older version of the manuscript. L.H.F. is the Herbaceous Diversity Network PI.

## Additional information

**Competing interests:** The authors declare no competing interests.

