## [Peer Review File · Nature Communications]

Reviewers' comments:

Reviewer #1 (Remarks to the Author):

Review of Wang et al. for Nature Communications

Wang presents analyses of species richness x biomass relationships and finds that the relationships change from positive to weakly positive to negative as the level of stress is reduced. This relationship was expected during the early days of the biodiversity-ecosystem functioning era (e.g. Figure 4 of Loreau et al. 2001 Science 294:806), but it has not really been tested in such a rigorous, comprehensive fashion. The authors find less variability across plots in species-rich situations, a result that was predicted from theory (Figure 4, Loreau et al. 2001). The data seems to be carefully collected, and I think it will make a valuable contribution to the literature on this topic. The statistical analyses are appropriate.

However, many details were left out, possibly due to the short format of Nature Communications. I have several comments that are important in addressing:

1. Title and throughout. I would not call this stability, but something like 'spatial stability'. I kept thinking of temporal variance in biomass production when thinking of stability, and I think most of your readers will as well. I think the paper and conclusions are much stronger if you are talking about this newly emerging concept of "spatial stability", which the authors have developed quite well in other papers.

Figure 1b. Y should be spatial stability.

2. Methods. Key details are missing. For example, when was biomass sampled? Was it peak biomass? If not, then it is standing crop biomass which is not the same thing as productivity. Peak biomass is a surrogate measure of productivity. If it is measured at peak, then this should be stated clearly.

I would also include the types of grasslands sampled in the methods. This was not very clear. Were deserts sampled? Were marshes sampled? It states in the discussion that arid, semi-arid and subhumid sites were sampled, but no humid grasslands or marshes? I would be very specific on this point in the methods.

3. Other smaller issues:

Line 147. "Synergy to antagonism" in what? This was vague and should be rewritten. Do you mean in competitive interactions or all interactions, or what?

Line 150 requires citations.

4. Lines 200-209. I agree that allelopathy might be involved with weak relationships between peak biomass and species richness, and I agree that this is an interesting idea. However, this has not been supported. An alternative explanation is that a priority effect due to species growing at different times explains this relationship. If low yielding species grow early in the year and suppress higher yielding species that grow later, it could lead to a reduced (or negative) complementarity effect. This could continue year after year, if the lower yielding species has an earlier phenology than the higher yielding species, and it suppresses growth of the higher yielding species. Much support has been found for this latter mechanism. It is true that high yielding species are not always the most dominant. I would read the literature on this and present several possible explanations behind the results, rather than just allelopathy.

It seems like Isbell et al. (2011) is appropriate to cite in this paper, especially when environmental changes are considered.

Reviewer #2 (Remarks to the Author):

The authors present a study that assesses whether the relationship between diversity and productivity or stability using 30 grassland sites across different regions worldwide. Using

structural equation modelling and bayesian models, they tested (i) whether the positive effects of plant diversity on plant biomass decreases as productivity increases, and (ii) whether the stability in plant biomass increases with increasing plant species richness. They found that plant diversity influences positively plant biomass at intermediate levels (as already published in Fraser et al. 2015 – Science) and that plant diversity contributes to the stability of plant biomass (calculated as SD and CV). They concluded that diversity stabilizes rather than increases biomass production at large spatial scales in natural grasslands. Because of an original approach using the dataset from HerbDivNet, to me this is an interesting study that warrants publication in Nature Communications. However, the definitions and results are sometimes unclear and incomplete for the reader and although the quality of science is good, the authors must add some precisions to be as clear as possible on their findings and their potential implications. I also suggest a slight reorganization of the discussion according to the main advancements of this study to improve the impact of this publication in Nature Communications. Because of these issues, coupled to some minor points, I recommend to the editor 'Revisions'.

General comment:

The authors present an experiment that addresses fundamental questions in ecosystem ecology and biodiversity which should be of interest to a wide audience. Overall, the manuscript is quite concise, well written and presents some interesting results for the understanding of plant diversity-stability. While I am generally positive about the manuscript, I will discuss two or three topics at length that need to be considered. First, the original definition of stability in ecology is the tendency to remain in, or return to, an equilibrium state (resistance vs resilience). Although many authors have used the term stability as the temporal or spatial invariance (often as $1/CV$), the original definition imply a response to a perturbation. Temporal or spatial variability may certainly be a reflection of responses to perturbations, but it may also reflect intrinsic dynamics that lead to population cycles or other complex temporal or spatial variation in the absence of perturbations. In other word, although plant diversity generally decreased variability in plant biomass, does plant diversity increases stability? Because the term stability is central in this manuscript, I think that a clear definition is required.

My main regret with this manuscript is the lack of soil physicochemical parameters in the different models. For instance, Kardol et al. (2018 - Nature) have recently found that species diversity increased temporal invariability (or stability in the context of this study), and these effects were greatest on the most productive and most fertile ecosystems. Because the authors used three categories in this study (low, medium, high biomass) for testing the 'stress-gradient hypothesis', it would be great to justify this gradient using both climatic AND soil factors. Further, it may improve model quality and help identify the mechanisms underlying the biodiversity effects (e.g., complementarity in resource use). After reading twice the manuscript, I recognize that the authors are open with the main limitations of their study, i.e., L172-L174, "We acknowledge that in addition to light, temperature and water, other abiotic and biotic factors also have important effects on plant richness and biomass.". However, I'm not convinced by their justification L177-179: "vegetation status on a global scale is mainly determined by climate, because soil conditions are also the result of combined effects of rock parent material, climate and biology." Vegetation is not only determined by climate at large spatial scales and this oversimplification made me blink a few times.

I have also some concerns about the results. I did not understand why they used a second structural equation modelling in their Figure 3 (i.e., Fig. 3b, plant biomass on plant richness). Based on the concepts and the ideas developed in the introduction section and in the conceptual model (i.e., stability or productivity vs species richness in Figure 1), this choice is confusing. Even if this result is briefly discussed in the discussion section (L171-L172), why the authors did not chose a double arrow (co-variation) if they think that 'there is a mutual causal relationship between aboveground live biomass and plant richness'. Do not get me wrong, I'm not saying that plant biomass cannot influence plant richness. But, although the first model is expected and logic based on the introduction section, I had to read several times the manuscript to understand what was the objective with the second panel of this figure.

In the Figure 6, the authors presented the relationship between variability (SD or CV) and species richness. However, they showed only the results across plots with the same richness. I would like to see the results for each group separately (low, medium, high biomass), at least in the supplementary data. I think that it would help to visualize whether this relationship vary with the

'stress-gradient hypothesis' because it is possible that this trend is only due to the group 'high biomass'. In my opinion, this point is very important and it would provide more information for the discussion section. Further, it will allow to test if the conclusions are robust or if they depend of the level of productivity.

In the Figure 4, the authors presented the partial effect of richness on biomass in the right corner of the main figure. I think that this sub panel is both clear and interesting. However, they repeated the model analysis for the three groups separately (L470-L472). I wonder why the authors did not use a multi-group modeling (e.g., <http://lavaan.ugent.be/tutorial/groups.html>). It would be great to visualize a model considering the three groups (low, medium, high biomass) at the same time and check if the results are similar (or not) if we consider all the groups independently or together.

Finally, in the discussion section, I expected more about the novelties of this study, i.e., the stability-species richness relationship, how it varies with the different groups of biomass, and the implications of these findings for ecosystem functioning in the context of global change. Instead, I found that the authors spend most of their discussion about the negative effects of biodiversity at high levels of productivity (L198-L210), or the potential mechanisms explaining biodiversity effects on productivity or ecosystem functioning (L180-197) (already discussed partially in some publications written by some of these authors). Further, the plan in the discussion section do not follow the hypotheses developed in the introduction, i.e., (i) the positive effects of plant diversity on plant biomass decreases as productivity increases, and then (ii) whether the stability in plant biomass increases with increasing plant species richness. Although the different paragraphs in the discussion section are very interesting (but sometimes fairly speculative), I believe that a re-structuring of the discussion according the main advancements of this study will improve the quality and clarity of this manuscript. In other words, I would re-use the same structure than presented in the introduction (richness vs productivity, then richness vs stability) while focusing more on the 'stability' aspect, the mechanisms underlying these stability, whether it varies between low and high biomass, and what are the implications for ecosystem functioning. I appreciated the ideas and concepts developed in the discussion, but I would prefer a discussion centered on the title of this publication in order to avoid misleading advertising, i.e., 'Global evidence of positive biodiversity effects on ecosystem stability in natural grasslands'.

Let me here reiterate that this is a nice study, of broad interest on an important topic. The findings of this experiment lead to interesting questions for future studies and especially concerning the emergence of new theories based on the stability-species richness relationship. Some details must be added to be as clear as possible and I recommend a slight reorganization of the discussion according to the main advancements of this study to improve the impact of this publication in Nature Communications. I look forward to the author's consideration and responses to the comments on their primary results.

Additional notes: Figure 2 can be moved in the supplementary data because it is only cited in the method section at the end of the manuscript (and thus does not appear in second position) and it is not essential for our understanding of the study. Further, is the correlation line in Figure 5 significant? Additional test must be provided to the readers.

Reviewer #3 (Remarks to the Author):

Wang et al. present a compelling analysis on the relationship between biodiversity and ecosystem productivity. Using a global dataset, they find that species richness has a positive effect on productivity in low productive environments but shifts to a negative effect in highly productive environments. They thus conclude that richness has a stabilizing effect on productivity. The global spatial extent of the dataset makes the results of this analysis extremely interesting. The first paragraph of the paper provides an excellent and clear summary of the results and their importance. I don't see any fatal flaws with the study but there are a number of important details that seem to be missing from the manuscript (e.g., spatial scale for analysis, equations for the structural equation models; see comments below). I realize that the word limit for articles is quite low in Nature Communications so some of this text may need to be incorporated into supplementary material if it won't fit in the methods section.

Major comments

1. Mismatch between the introduction and the discussion – Early in the manuscript the authors state that most biodiversity-productivity relationships are positive but that there is variation in the strength and shape of these relationships (lines 19-20, 32-36). However, the discussion section makes claims that the literature is pretty conclusive on the positive relationship (line 138-139) and that the current research debunks this thinking. Thus, the set up in the introduction does not seem to match the discussion section. Additionally, stress is discussed quite a bit in the intro but issues of stress are not addressed in the results or revisited in the discussion. Overall, there seems to be a disconnect between the ideas in the intro and those in the discussion.

2. Clarity in terminology – The terminology is not clear in many places and in some cases the jargon muddles the meaning. For example, the list of possible mechanisms to explain relationships between richness and productivity (line 38-39) is not useful without further explanation and the conditions under which these mechanisms are likely to be relevant. Although it becomes clear what the authors mean by the terms biodiversity (species richness) and productivity (plant biomass) in the last paragraph of the intro, it would help the reader to define these terms upfront. Also, I believe that live plant biomass is used as a surrogate for productivity, but this is not clear. Plant growth is not measured or used in the analysis, correct? Overall please review the language throughout. Related, given that “less than 10% of the variance in biomass (or richness) could be explained by number of daylight hours, temperature, precipitation and richness (or biomass)” (line 94-96) does this suggest that richness is a relatively minor piece of the puzzle?

3. Treatment of space in the paper – I find the treatment of space in the paper confusing. It seems that there are two issues of space that are of concern in this manuscript:

-The first is whether the relationship between plant richness and biomass holds across spatial locations around the world. Presumably, this is assessed by comparing the same size plots across sampling sites (e.g., the 30 grassland fields).

-The second issue is whether the relationship between plant richness and biomass holds across spatial extends. Presumably, this is assessed by comparing richness/biomass in areas of varying size (e.g., collecting data at various plot/grid sizes within a single or multiple locations).

Both of these issues are brought up, and somewhat conflated, in the introduction and in the discussion. I think that this paper is primarily focused on the first issue. However, the lack of details on the data structure and models makes this unclear. More on that below.

4. Incomplete description of the survey data – Although the paper states that the results are based on 9,640 plots in 151 grids in 30 natural grasslands, it is not clear what is the spatial unit of analysis. Are the authors comparing between productivity among 30 grassland habitats, using the 9,640 plots as pseudo-replicates? How do the grids fit in? It seems like the 30 grasslands is the appropriate unit of analysis (e.g., Fig 2; as plots are pseudo-replicates) but there are clearly more points in Figs 4-6.

5. Incomplete description of the models – The authors should include information on their structural equation models. What are the equations for those models? How are they used to infer causal mechanisms? There are no real details on the structural equation models so it is difficult to evaluate the implementation of this approach (and I am skeptical that they can be used to infer causality based on the data structure. I don't see this as a major flaw but suggest perhaps some rephrasing). Additionally, it is not clear why the authors choose to use both structural equation models and Bayesian hierarchical models. What unique pieces of information do these two analyses provide? The results suggest that these two modeling approaches provide the same information. The description of the Bayesian model is comparatively more transparent, although the k index is not clear (e.g., what is a richness level?). Also ϵ_k is used in equation 2 but not defined in the model (equation 1). The authors should consider including the code used for the analysis as supplementary material, which could help clarify the various approaches, including the priors used for the Bayesian analysis (an important detail that does not seem to be in the methods).

Reviewers' comments:

Reviewer #1 (Remarks to the Author):

Review of Wang et al. for Nature Communications

Q1:

Wang presents analyses of species richness x biomass relationships and finds that the relationships change from positive to weakly positive to negative as the level of stress is reduced. This relationship was expected during the early days of the biodiversity-ecosystem functioning era (e.g. Figure 4 of Loreau et al. 2001 Science 294:806), but it has not really been tested in such a rigorous, comprehensive fashion. The authors find less variability across plots in species-rich situations, a result that was predicted from theory (Figure 4, Loreau et al. 2001). The data seems to be carefully collected, and I think it will make a valuable contribution to the literature on this topic. The statistical analyses are appropriate.

Response: Yes, the idea of this study was inspired by Loreau et al. (2001). Thank you very much for these positive comments.

However, many details were left out, possibly due to the short format of Nature Communications. I have several comments that are important in addressing:

Q2:

1. Title and throughout. I would not call this stability, but something like 'spatial stability'. I kept thinking of temporal variance in biomass production when thinking of stability, and I think most of your readers will as well. I think the paper and conclusions are much stronger if you are talking about this newly emerging concept of "spatial stability", which the authors have developed quite well in other papers.

Figure 1b. Y should be spatial stability.

Response: We agree with this comment, the term "stability" has been changed to "spatial stability" throughout the revised manuscript, including Title and Figure 1b. We have also supplemented the definition of 'spatial stability' in the Introduction, where we say: 'Stability has many different definitions in ecology (Pimm 1984; Loreau et al. 2002). Here we focus on spatial stability, S , which measures the similarity of ecosystem properties such as productivity across different grassland communities worldwide. S is defined as $1/\text{variability}$, where variability is a measure of the magnitude of spatial changes in an ecosystem property.'

Q3:

2. Methods. Key details are missing. For example, when was biomass sampled? Was it peak biomass? If not, then it is standing crop biomass which is not the same thing as productivity. Peak biomass is a surrogate measure of productivity. If it is measured at peak, then this should be stated clearly.

I would also include the types of grasslands sampled in the methods. This was not very clear. Were deserts sampled? Were marshes sampled? It states in the discussion that arid, semi-arid and subhumid sites were sampled, but no humid grasslands or marshes? I would be very specific on this point in the methods.

Response: Yes, the aboveground live biomass was peak biomass, we have made this clear in the Introduction and Methods. We have also added the types of grasslands sampled in the Introduction and Methods, where we say: ‘Grassland type was assigned to 5 categories: temperate, temperate wet meadow, Mediterranean, tropical and subtropical, and alpine’.

Q4:

3. Other smaller issues:

Line 147. “Synergy to antagonism” in what? This was vague and should be rewritten. Do you mean in competitive interactions or all interactions, or what?

Line 150 requires citations.

Response: We agree, and have reworded the sentence. It now reads ‘the variations in the sign and strength of interspecific interactions along the stress gradient’. We added two references (Isbell et al. 2011; Lewandowsky 2011) to the original line 150.

Q5:

4. Lines 200-209. I agree that allelopathy might be involved with weak relationships between peak biomass and species richness, and I agree that this is an interesting idea. However, this has not been supported. An alternative explanation is that a priority effect due to species growing at different times explains this relationship. If low yielding species grow early in the year and suppress higher yielding species that grow later, it could lead to a reduced (or negative) complementarity effect. This could continue year after year, if the lower yielding species has an earlier phenology than the higher yielding species, and it suppresses growth of the higher yielding species. Much support has been found for this latter mechanism. It is true that high yielding species are not always the most dominant. I would read the literature on this and present several possible explanations behind the results, rather than just allelopathy.

Response: We thank the Reviewer for this thoughtful comment. The priority effect has been discussed in the Discussion. We have also supplemented the relevant references.

Q6:

It seems like Isbell et al. (2011) is appropriate to cite in this paper, especially when environmental changes are considered.

Response: We agree, Isbell et al. (2011) has been cited in our revised manuscript.

Reviewer #2 (Remarks to the Author):

Q7:

The authors present a study that assesses whether the relationship between diversity and productivity or stability using 30 grassland sites across different regions worldwide. Using structural equation modelling and bayesian models, they tested (i) whether the positive effects of plant diversity on plant biomass decreases as productivity increases, and (ii) whether the stability in plant biomass increases with increasing plant species richness. They found that plant diversity influences positively plant biomass at intermediate levels (as already published in Fraser et al. 2015 – Science)

and that plant diversity contributes to the stability of plant biomass (calculated as SD and CV). They concluded that diversity stabilizes rather than increases biomass production at large spatial scales in natural grasslands. Because of an original approach using the dataset from HerbDivNet, to me this is an interesting study that warrants publication in Nature Communications. However, the definitions and results are sometimes unclear and incomplete for the reader and although the quality of science is good, the authors must add some precisions to be as clear as possible on their findings and their potential implications. I also suggest a slight reorganization of the discussion according to the main advancements of this study to improve the impact of this publication in Nature Communications. Because of these issues, coupled to some minor points, I recommend to the editor 'Revisions'.

Response: We are very grateful for these helpful comments. Our manuscript has been revised and supplemented, including the addition of a definition of spatial stability, some further analyses and their potential implications.

General comment:

Q8:

The authors present an experiment that addresses fundamental questions in ecosystem ecology and biodiversity which should be of interest to a wide audience. Overall, the manuscript is quite concise, well written and presents some interesting results for the understanding of plant diversity-stability. While I am generally positive about the manuscript, I will discuss two or three topics at length that need to be considered. First, the original definition of stability in ecology is the tendency to remain in, or return to, an equilibrium state (resistance vs resilience). Although many authors have used the term stability as the temporal or spatial invariance (often as $1/CV$), the original definition imply a response to a perturbation. Temporal or spatial variability may certainly be a reflection of responses to perturbations, but it may also reflect intrinsic dynamics that lead to population cycles or other complex temporal or spatial variation in the absence of perturbations. In other word, although plant diversity generally decreased variability in plant biomass, does plant diversity increases stability? Because the term stability is central in this manuscript, I think that a clear definition is required.

Response: We agree with this comment and have supplemented the definition of stability in the Introduction, where we say: 'Stability has many different definitions in ecology (Pimm 1984; Loreau et al. 2002). Here we focus on spatial stability, S , which measures the similarity of ecosystem properties such as productivity across different grassland communities worldwide. S is defined as $1/\text{variability}$, where variability is a measure of the magnitude of spatial changes in an ecosystem property.'

Q9:

My main regret with this manuscript is the lack of soil physicochemical parameters in the different models. For instance, Kardol et al. (2018 - Nature) have recently found that species diversity increased temporal invariability (or stability in the context of this study), and these effects were greatest on the most productive and most fertile ecosystems. Because the authors used three categories in this study (low, medium, high biomass) for testing the 'stress-gradient hypothesis', it would be great to justify this gradient using both climatic AND soil factors. Further, it may improve model quality and help identify the mechanisms underlying the biodiversity effects (e.g.,

complementarity in resource use). After reading twice the manuscript, I recognize that the authors are open with the main limitations of their study, i.e., L172-L174, “We acknowledge that in addition to light, temperature and water, other abiotic and biotic factors also have important effects on plant richness and biomass.” However, I’m not convinced by their justification L177-179: “vegetation status on a global scale is mainly determined by climate, because soil conditions are also the result of combined effects of rock parent material, climate and biology.” Vegetation is not only determined by climate at large spatial scales and this oversimplification made me blink a few times.

Response: We agree that soil factors are important and that climate factors explain only a small part of biomass variation. We have deleted our previous assertion, and now cite Kardol et al. (2018) as their results are consistent with, and complementary to, those of our new analysis in biomass gradient (see Q11). We are grateful to the Reviewer for providing us with this reference.

Q10:

I have also some concerns about the results. I did not understand why they used a second structural equation modelling in their Figure 3 (i.e., Fig. 3b, plant biomass on plant richness). Based on the concepts and the ideas developed in the introduction section and in the conceptual model (i.e., stability or productivity vs species richness in Figure 1), this choice is confusing. Even if this result is briefly discussed in the discussion section (L171-L172), why the authors did not chose a double arrow (co-variation) if they think that ‘there is a mutual causal relationship between aboveground live biomass and plant richness’. Do not get me wrong, I’m not saying that plant biomass cannot influence plant richness. But, although the first model is expected and logic based on the introduction section, I had to read several times the manuscript to understand what was the objective with the second panel of this figure.

Response: We agree that presenting two models is somewhat confusing, we have deleted Fig. 3b and the associated Supplementary Table 1, and modified the result accordingly, but this idea is still present in the discussion.

Q11:

In the Figure 6, the authors presented the relationship between variability (SD or CV) and species richness. However, they showed only the results across plots with the same richness. I would like to see the results for each group separately (low, medium, high biomass), at least in the supplementary data. I think that it would help to visualize whether this relationship vary with the ‘stress-gradient hypothesis’ because it is possible that this trend is only due to the group ‘high biomass’. In my opinion, this point is very important and it would provide more information for the discussion section. Further, it will allow to test if the conclusions are robust or if they depend of the level of productivity.

Response: We thank the Reviewer for this thoughtful comment. The 151 grids were divided into three equal groups (low, medium, and high biomass) depending on their mean live biomass as that of Fig. 3, we then repeated the simple regression analysis and the hierarchical Bayesian model for each of the three groups separately (results are shown in Supplementary Figures 2 and 3). We found that although the negative correlation between species richness and spatial variability exists generally, the higher the productivity level, the more significant

the negative correlation. It is true that ‘high biomass’ contributes the most to this relationship. Both ‘Results’ and ‘Discussion’ sections are supplemented accordingly.

Q12:

In the Figure 4, the authors presented the partial effect of richness on biomass in the right corner of the main figure. I think that this sub panel is both clear and interesting. However, they repeated the model analysis for the three groups separately (L470-L472). I wonder why the authors did not use a multi-group modeling (e.g., <http://lavaan.ugent.be/tutorial/groups.html>). It would be great to visualize a model considering the three groups (low, medium, high biomass) at the same time and check if the results are similar (or not) if we consider all the groups independently or together.

Response: We agree with this comment. We have used a multi-group modeling to analyze the three groups (low, medium, and high biomass) at the same time. We found that the results are similar to those of the separate analysis. The results of both analyses are presented in the main text.

Q13:

Finally, in the discussion section, I expected more about the novelties of this study, i.e., the stability-species richness relationship, how it varies with the different groups of biomass, and the implications of these findings for ecosystem functioning in the context of global change. Instead, I found that the authors spend most of their discussion about the negative effects of biodiversity at high levels of productivity (L198-L210), or the potential mechanisms explaining biodiversity effects on productivity or ecosystem functioning (L180-197) (already discussed partially in some publications written by some of these authors). Further, the plan in the discussion section do not follow the hypotheses developed in the introduction, i.e., (i) the positive effects of plant diversity on plant biomass decreases as productivity increases, and then (ii) whether the stability in plant biomass increases with increasing plant species richness. Although the different paragraphs in the discussion section are very interesting (but sometimes fairly speculative), I believe that a re-structuration of the discussion according the main advancements of this study will improve the quality and clarity of this manuscript. In other words, I would re-use the same structure than presented in the introduction (richness vs productivity, then richness vs stability) while focusing more on the ‘stability’ aspect, the mechanisms underlying these stability, whether it varies between low and high biomass, and what are the implications for ecosystem functioning. I appreciated the ideas and concepts developed in the discussion, but I would prefer a discussion centered on the title of this publication in order to avoid misleading advertising, i.e., ‘Global evidence of positive biodiversity effects on ecosystem stability in natural grasslands’.

Response: Thank you for pointing this out. We have revised the Discussion section accordingly. We added a paragraph to discuss the variation of the relationship between species richness and spatial variability along a biomass gradient, and illustrate the significance of this result in the context of global change. In fact, the reason why we discuss the negative correlation between biodiversity and productivity in high-productivity environment is to explain the mechanism of spatial stability. We have made a supplementary statement in the Discussion section, where we say: “With the decrease of environmental stress (that is, as productivity increases), the positive correlation between biodiversity and productivity weakens gradually (Fig. 1a), and eventually becomes negative. This gradual change explains why spatial stability

increases along the productivity gradient. Existing theories, however, do not fully explain the negative relationship between biodiversity and productivity.”

Q14:

Let me here reiterate that this is a nice study, of broad interest on an important topic. The findings of this experiment lead to interesting questions for future studies and especially concerning the emergence of new theories based on the stability-species richness relationship. Some details must be added to be as clear as possible and I recommend a slight reorganization of the discussion according to the main advancements of this study to improve the impact of this publication in Nature Communications. I look forward to the author’s consideration and responses to the comments on their primary results.

Response: Thank you very much for these helpful comments.

Q15:

Additional notes: Figure 2 can be moved in the supplementary data because it is only cited in the method section at the end of the manuscript (and thus does not appear in second position) and it is not essential for our understanding of the study. Further, is the correlation line in Figure 5 significant? Additional test must be provided to the readers.

Response: Figure 2 has been moved to the supplementary data (Supplementary Figure 1). After removing Figure 2, the original Figure 5 becomes Figure 4, and the correlation line in this figure is significant. We have added the test statistics in the figure legend, “The solid line is the estimated relationship between mean biomass and diversity effect. The average slope of the line is -0.033 (95% CI: [-0.041, -0.024]).” We have also updated the analysis and result associated with Figure 4 about the shift of the diversity effects across the productivity gradient. At the first level of our Bayesian model, we fitted the relationship between plot-level richness and biomass (Equation 1a and supplementary code). We included grid-associated random effects for both the intercept and slope (alphas 0 and 1). We can interpret this grid-level intercept as the estimated grid-level average biomass. In the earlier version of the analysis, we further fitted the relationship between the grid-level intercept (i.e., estimated grid-level average biomass) and the observed grid-level average biomass. We think it is logically circular to regress biomass against biomass. Therefore, in the current version, we fitted the grid-level intercept with the grid-associated random effect only (Equation 1d and supplementary code). The updated analysis did not change our conclusion.

Reviewer #3 (Remarks to the Author):

Q16:

Wang et al. present a compelling analysis on the relationship between biodiversity and ecosystem productivity. Using a global dataset, they find that species richness has a positive effect on productivity in low productive environments but shifts to a negative effect in highly productive environments. They thus conclude that richness has a stabilizing effect on productivity. The global spatial extent of the dataset makes the results of this analysis extremely interesting. The first

paragraph of the paper provides an excellent and clear summary of the results and their importance. I don't see any fatal flaws with the study but there are a number of important details that seem to be missing from the manuscript (e.g., spatial scale for analysis, equations for the structural equation models; see comments below). I realize that the word limit for articles is quite low in Nature Communications so some of this text may need to be incorporated into supplementary material if it won't fit in the methods section.

Response: We are very grateful for these helpful comments. According to your comments, our manuscript have been revised and supplemented accordingly.

Major comments

Q17:

1. Mismatch between the introduction and the discussion – Early in the manuscript the authors state that most biodiversity-productivity relationships are positive but that there is variation in the strength and shape of these relationships (lines 19-20, 32-36). However, the discussion section makes claims that the literature is pretty conclusive on the positive relationship (line 138-139) and that the current research debunks this thinking. Thus, the set up in the introduction does not seem to match the discussion section. Additionally, stress is discussed quite a bit in the intro but issues of stress are not addressed in the results or revisited in the discussion. Overall, there seems to be a disconnect between the ideas in the intro and those in the discussion.

Response: We thank the Reviewer for pointing this out. We have modified and supplemented the Discussion to match the Introduction. Also, 'stress' is revisited in the discussion.

Q18:

2. Clarity in terminology – The terminology is not clear in many places and in some cases the jargon muddles the meaning. For example, the list of possible mechanisms to explain relationships between richness and productivity (line 38-39) is not useful without further explanation and the conditions under which these mechanisms are likely to be relevant. Although it becomes clear what the authors mean by the terms biodiversity (species richness) and productivity (plant biomass) in the last paragraph of the intro, it would help the reader to define these terms upfront. Also, I believe that live plant biomass is used as a surrogate for productivity, but this is not clear. Plant growth is not measured or used in the analysis, correct? Overall please review the language throughout. Related, given that “less than 10% of the variance in biomass (or richness) could be explained by number of daylight hours, temperature, precipitation and richness (or biomass)” (line 94-96) does this suggest that richness is a relatively minor piece of the puzzle?

Response: We have now deleted the list of the possible mechanisms to explain relationships between richness and productivity, because our paper is not suitable for a detailed introduction to these mechanisms. We have now defined 'biodiversity' and 'productivity' in the introduction. Yes, plant growth was not measured in HerbDivNet data, so we used live plant biomass at peak biomass as a surrogate for primary productivity as many previous studies did. Indeed, at the global extent, according to our research, richness did not have a general and strong correlation with biomass. Plot-level richness-biomass relationships actually varied greatly according to the grid-level productivity gradient. This variation made

richness a strong predictor for spatial stability of biomass at the global scale, which is consistent with our hypotheses.

Q19:

3. Treatment of space in the paper – I find the treatment of space in the paper confusing. It seems that there are two issues of space that are of concern in this manuscript:

-The first is whether the relationship between plant richness and biomass holds across spatial locations around the world. Presumably, this is assessed by comparing the same size plots across sampling sites (e.g., the 30 grassland fields).

-The second issue is whether the relationship between plant richness and biomass holds across spatial extends. Presumably, this is assessed by comparing richness/biomass in areas of varying size (e.g., collecting data at various plot/grid sizes within a single or multiple locations).

Both of these issues are brought up, and somewhat conflated, in the introduction and in the discussion. I think that this paper is primarily focused on the first issue. However, the lack of details on the data structure and models makes this unclear. More on that below.

Response: Yes, the first relationship between richness and biomass is assessed by comparing the same size plots ($N = 9,640$ plots, each $1\text{ m} * 1\text{ m}$) across the 30 sampling sites, the results are now shown in Figs 2 and 3. The climate conditions of different sites are different. Within the same site, there are significant productivity gradients between grids (Fraser et al. 2015). That is, in the same site, even though the climatic conditions are similar, the soil conditions of different grids are different, which means that ‘site’ is not the appropriate unit of analysis. So we then chose ‘grid’ as an analytical unit to explore the relationship between richness and biomass, because each grid was established in a patch of relatively homogeneous area, which means that each grid is small enough to exclude significant variations in abiotic conditions. This allowed us to test the effects of biodiversity on productivity. We fitted grid as a random effect in the Bayesian model to control potential non-independence among plots within a grid. This result is now shown in Fig. 4. Each spatial unit (plot, grid, site, plot group, grid group, or richness group) used in the analysis has been clearly described in the revised manuscript.

Q20:

4. Incomplete description of the survey data – Although the paper states that the results are based on 9,640 plots in 151 grids in 30 natural grasslands, it is not clear what is the spatial unit of analysis. Are the authors comparing between productivity among 30 grassland habitats, using the 9,640 plots as pseudo-replicates? How do the grids fit in? It seems like the 30 grasslands is the appropriate unit of analysis (e.g., Fig 2; as plots are pseudo-replicates) but there are clearly more points in Figs 4-6.

Response: Because there is a clear productivity gradient between the different grids within each site (Fraser et al. 2015), so ‘site’ is not the appropriate unit of analysis. We chose ‘grid’ as an analytical unit to explore the relationship between richness and biomass, because each grid was established in a patch of relatively homogeneous area. Each spatial unit (plot, grid, site, plot group, grid group, or richness group) used in the analysis has been clearly described in the revised manuscript.

Q21:

5. Incomplete description of the models – The authors should include information on their structural

equation models. What are the equations for those models? How are they used to infer causal mechanisms? There are no real details on the structural equation models so it is difficult to evaluate the implementation of this approach (and I am skeptical that they can be used to infer causality based on the data structure. I don't see this as a major flaw but suggest perhaps some rephrasing). Additionally, it is not clear why the authors choose to use both structural equation models and Bayesian hierarchical models. What unique pieces of information do these two analyses provide? The results suggest that these two modeling approaches provide the same information. The description of the Bayesian model is comparatively more transparent, although the k index is not clear (e.g., what is a richness level?). Also ϵ_k is used in equation 2 but not defined in the model (equation 1).

Response: We have supplemented the codes associated with structural equation models (SEM) and Bayesian models. We have also added more detailed description about the structural equation models in Methods. We have rephrased the texts associated with causality. We used both SEM and Bayesian models because they are complementary and each has its own advantages. The SEM can handle multiple causal relationships among various variables, which is common in complex observational data. The Bayesian model can handle data with complicated hierarchical structures (e.g., plot nested inside grid) or with unbalanced design (e.g., some species richness levels with very few replications). However, these complicated data structure would be difficult to implement in SEM. We have provided this clarification in the text, "The Bayesian model is complementary to the SEM, but has the advantages in avoiding categorization of the plots or grids into groups artificially." We have added the description about richness level k , "plots with the same richness value k were at the richness level k " and ϵ_k , " ϵ_k was the difference between observed and predicted $\log(\sigma_k)$."

Q22:

The authors should consider including the code used for the analysis as supplementary material, which could help clarify the various approaches, including the priors used for the Bayesian analysis (an important detail that does not seem to be in the methods).

Response: We have supplemented the R-code for the Bayesian analysis and structural equation models.

Reviewers' comments:

Reviewer #1 (Remarks to the Author):

The authors did a good job addressing the reviewers concerns.

Reviewer #2 (Remarks to the Author):

I would like to thank the authors for their consideration of the main critical points raised by the two other reviewers and myself during the first round of review. They considerably improved the manuscript and they did their best to address most of comments. Overall, I am supportive of this new version and I think that the manuscript can be accepted for publication in 'Nature Communications'.

Yet, the changes made in this revised version raised new minor comments. These new comments are not mandatory but they may improve the clarity and quality of the manuscript. For instance, I appreciated the new Figure 2 that shows clearly the effects of biodiversity on productivity. But, I would like to know if the negative impact of biodiversity on productivity vary with the three levels of productivity (low, medium and high) in hierarchical Bayesian models (Figure 4)? The authors presented the different levels of productivity in Figure 3 (SEM per group) but they did not use the same approach in Figure 4, in which they present only the general model. I may have missed something, but would it be possible to visualize in Bayesian models 'the sign and strength of the biodiversity effects' for each level of productivity, at least in Supplementary data. This would help the reader to assess if the results are robust when using different approaches, especially because the three levels of productivity are central in the context of this manuscript.

Further, I think that it is useful to show first the relationship between biodiversity and productivity in order to focus on what is new and important in this manuscript: 'the relationship between biodiversity and stability'. Here, the authors have done a fantastic job and addressed most of my comments with Figure S2 and S3. I think that these figures improve considerably the clarity of the manuscript. However, they did not discuss about the differences between SD and CV, especially for the 'low productivity' group (Fig S2c/S2f – S3c/S3f). Here the trends are different and the relationship not always significant. Further, when observing the results for SD, the relationship between biodiversity and spatial variability seems fairly negative in the 'high productivity' group, almost null in the 'medium productivity' group, and non-significant for the 'low productivity' group. This was not the case for CV. Thus, my questions are simple: is this important for data interpretation? What are the implications of these findings? Although I appreciate that the authors discussed the changes in the relationship between species diversity and stability across the productivity gradient in their new version of the discussion section (i.e., higher on the most productive ecosystems) (L187-L194), it would be helpful for the readers to comment briefly the differences in the results between CV and SD, at least in the Supplementary material. I recognize that the trends are overall similar, but the results are not identical, especially because the relationships between diversity and stability for the 'medium productivity' group the 'low productivity' group present relatively similar slopes for CV, contrary to SD.

Finally, I think that the conclusions of this publication are pretty vague and the perspectives inexistent. Do not get me wrong, I believe that the last paragraph is well written and interesting. But, the two first sentences are generic, and the implications for the audience relatively limited. I suggest to summarize the results and then to extend their conclusions to a broader public in order to increase the impact of this publication in Nature Communications. What are the implication of their findings for conservation biology or ecosystem functioning? What is the main message (beyond the results) that the readership needs to remember?

Overall, let me reiterate that the authors satisfied most of my comments and considerably improved the clarity of their manuscript. I believe that this publication is outstanding in its new version and if the two other reviewers agree with the revisions, I would support this work to be published in Nature Communications, which should be of interest to a wide audience.

N. Fanin

Reviewer #4 (Remarks to the Author) (Editor's note - Reviewer recruited to comment on behalf of Reviewer #3 regarding the Bayesian approaches):

I enjoyed reviewing this paper and felt the improvements made by the authors in this revision added a lot of clarification. My comments pertain to the details in the Bayesian model and while they are somewhat extensive, I hope the authors find them helpful in terms of improving the manuscript.

The authors posit that biodiversity stabilizes plant productivity across space due to the positive effects of species richness on productivity in harsh environments giving way to constant and finally negative effects as stress decreases (Fig 1 and lines 50-70). In their Bayesian model, these changing biodiversity effects across stress gradients (represented by average grid productivity) are nicely captured using random coefficients where slope is a function of grid productivity (a proxy for the stress all plots within a grid are assumed to be subject to). If spatial stability is in fact occurring due to the mechanism the authors propose, then the reduced variation in productivity seen in species rich plots is due mainly to this convergence of predicted grid productivity as biodiversity increases (this interpretation is taken directly from Fig 1 and lines 50-70).

Based on their Bayesian model, the authors use two key parameter estimates to support the conceptual model visualized in Fig 1. The first is that the γ_1 is negative resulting in the $\alpha_{i,j}$ slope parameters become increasingly negative as grid productivity increases and the second is that within grid plot-level variation decreases as biodiversity increases. While this probably results in convergence resulting in spatial stability across a biodiversity gradient, it is hard to know from the way the authors have chosen to present their results just how much global variation in plot-level productivity varies with respect to biodiversity, as this depends heavily on the combination of intercept (α_{0j}) and slope (α_{1j}), and range of species richness values for each of the grid productivity gradients. To address this, I suggest the authors use the output from their Bayesian model to do two things: 1) recreate Fig. 1a using the grid-level slope and intercepts posterior means so we can see visually whether your statistical model actually supports the convergence visualized in Fig. 1a and 2) recreate Fig. 1b by predicting plant productivity for equal number of plots from each grid along this biodiversity gradient, computing the spatial stability amongst these plots as a derived quantity in your Bayesian model, and then plotting stability as a function of biodiversity.

I have the following comments regarding the Bayesian model presented in the methods section:

1) It is standard practice to perform posterior predictive checks (see Gelman and Hill, 2007 or Hobbs and Hooten, 2015) to assess model fit. If the authors haven't already done these then they should, especially since their model is being used mainly to variation in productivity. These checks are easy to do computationally and asking the authors to report these are not onerous.

2) In addition to the code, the authors should provide summaries of the model output in a supplement.

3) The R^2 calculation (line 360) captures the amount of variation in logged SD or CV explained by species richness but is not a good metric of overall model fit. Again, I would do posterior predictive checks for this. Also, equation 1c is incorrect as $\alpha_{0,j}$ is modeled hierarchically in your R code. I believe this epsilon should be subscripted with a k and is actually a derived quantity computed as $\log(\sigma_k) - (\beta_0 + \beta_1 * \text{richness}_k)$.

4) Equations 1a-1d do not match the R code as they are missing the random intercept model. Also $\alpha_{1,j}$ does = normal it is ~ normal. I really would present the model in these equations using a joint distribution, as described in Hobbs and Hooten 2015, but I understand if you leave it presented in this fashion (after correcting).

5) Productivity is strictly positive and given that you have values close to zero I would suggest using the lognormal or gamma distribution instead of the normal to model productivity instead. I don't think this would require you to change the distributions for the other parameters. Also, the biodiversity effects being modeled here ($\alpha_{1,j}$) are all additive making them hard to compare

across grid-level productivity gradients. If you used the lognormal or gamma distribution, it would be easy to use the log-link function to model these biodiversity effects as multiplicative, which are much easier to compare across productivity/stress gradients.

6) I am not clear why your model does not include other covariates from the SEM known to affect plot level productivity, such as daylight hours, temperature, or precipitation, as covariates in Equation 1d?

7) I agree with your choice of plot- and grid-level for this analysis (response to Q19/20). However, there is likely non-independence amongst the grids due to site effects and you should consider adding a random site effect to account for this.

8) In Fig. 1a there is a clear correlation between the slope and intercepts (more productive grids have flatter or more negative slopes but larger intercept values), but in your random coefficients model you do not model this correlation at all. Given that such a negative correlation btw slope and intercept is needed to get these biodiversity-productivity lines to converge as in Fig. 1a, modeling it would be helpful.

9) Lines 353 and 357: What do you mean by first and second hypothesis? Searching earlier in the text, I can't find a reference to two separate hypotheses being tested by the Bayesian model.

10) How is the Bayesian model described in lines 125-132 different from the Bayesian model described in lines 115-118, Figure 4, and extensively in the methods? If the Bayesian model present in Fig. 5b,d is the same as the one previously described earlier lines 115-118, then I would omit the simple linear regressions. If it is not, then I would omit the new Bayesian analysis.

Regarding questions raised by the Reviewer #3

Q17: Adequately addressed by the reviewers.

Q18: I had the same concern as Reviewer #3 when reading this manuscript (especially after seeing the proportion of variation explained in Fig. 2). The authors responded that because there was substantial variation in the grid-specific slope parameters and this "made richness a strong predictor for spatial stability of biomass at the global scale, which is consistent with our hypotheses." This might be true (see my main comment above) but the authors should compute Bayesian R^2 to see the proportion of plot-level productivity explained by grid-level richness effects to really address the reviewer's concern. Again, this would be straightforward to add to the model code.

Q19/Q20: I agree with the authors choice of grid- and plot level for this analysis. However, I share the concern of the previous reviewer, that since grids are nested within sites are likely non-independent and the authors should also add a site-level random effect.

Q21/Q22: Adequately addressed by the reviewers.

Reviewers' comments:

Reviewer #1 (Remarks to the Author):

Question-1:

The authors did a good job addressing the reviewers concerns.

Response: Thank you very much.

Reviewer #2 (Remarks to the Author):

Question-2:

I would like to thank the authors for their consideration of the main critical points raised by the two other reviewers and myself during the first round of review. They considerably improved the manuscript and they did their best to address most of comments. Overall, I am supportive of this new version and I think that the manuscript can be accepted for publication in 'Nature Communications'.

Response: We are very grateful for these helpful and positive comments.

Question-3:

Yet, the changes made in this revised version raised new minor comments. These new comments are not mandatory but they may improve the clarity and quality of the manuscript. For instance, I appreciated the new Figure 2 that shows clearly the effects of biodiversity on productivity. But, I would like to know if the negative impact of biodiversity on productivity vary with the three levels of productivity (low, medium and high) in hierarchical Bayesian models (Figure 4)? The authors presented the different levels of productivity in Figure 3 (SEM per group) but they did not use the same approach in Figure 4, in which they present only the general model. I may have missed something, but would it be possible to visualize in Bayesian models 'the sign and strength of the biodiversity effects' for each level of productivity, at least in Supplementary data. This would help the reader to assess if the results are robust when using different approaches, especially because the three levels of productivity are central in the context of this manuscript.

Response: We thank the reviewer for these excellent suggestions. We have added the biodiversity effects for the three levels of productivity (low, medium, and high) in hierarchical Bayesian models as an inset in Figure 5. The results are consistent with those from SEM (Figure 3).

Question-4:

Further, I think that it is useful to show first the relationship between biodiversity and productivity in order to focus on what is new and important in this manuscript: 'the relationship between biodiversity and stability'. Here, the authors have done a fantastic job and addressed most of my comments with Figure S2 and S3. I think that these figures improve considerably the clarity of the manuscript. However, they did not discuss about the differences between SD and CV, especially for the 'low productivity' group (Fig S2c/S2f – S3c/S3f). Here the trends are different and the

relationship not always significant. Further, when observing the results for SD, the relationship between biodiversity and spatial variability seems fairly negative in the ‘high productivity’ group, almost null in the ‘medium productivity’ group, and non-significant for the ‘low productivity’ group. This was not the case for CV. Thus, my questions are simple: is this important for data interpretation? What are the implications of these findings? Although I appreciate that the authors discussed the changes in the relationship between species diversity and stability across the productivity gradient in their new version of the discussion section (i.e., higher on the most productive ecosystems) (L187-L194), it would be helpful for the readers to comment briefly the differences in the results between CV and SD, at least in the Supplementary material. I recognize that the trends are overall similar, but the results are not identical, especially because the relationships between diversity and stability for the ‘medium productivity’ group the ‘low productivity’ group present relatively similar slopes for CV, contrary to SD.

Response: We agree with this comment. We first described the difference in the Results: ‘Although the trends were similar overall, the results were slightly different depending on whether variability was measured by SD or CV. The relationship between biodiversity and spatial variability was fairly negative in the ‘high productivity’ group, weakly negative in the ‘medium productivity’ group, and non-significant for the ‘low productivity’ group when using SD. When using CV, all the relationships were significantly negative.’

Then, in the Discussion section, we analyzed the possible reasons for this difference: ‘The relationships between richness and spatial variability had consistently negative slopes along the productivity gradients when using CV as a measure of variability, but this was not the case when using SD (see Supplementary Figures 2 and 3). This is due to the difference in the measurement of spatial variability: SD is a measure of the average deviation from the mean productivity, and is thus an absolute measure of variability; while CV is the ratio between SD and mean productivity, and is thus a relative measure that removes the impact of mean productivity. In the high productivity group, the relationship between richness and productivity was negative (see the insets in Figs 3 and 5), thus reducing the negative correlation between richness and spatial variability when the measure of variability is changed from SD to CV. There is still a fairly negative correlation between richness and CV, indicating that our result that biodiversity increases spatial stability is very robust in highly productive environments. However, as the level of productivity decreases (in medium and low productivity groups), the relationship between richness and productivity becomes positive, which contributes to strengthening the negative correlation between richness and variability when SD is replaced by CV. Since CV removes the impact of mean productivity, it is a more widely used measure of ecosystem variability (Tilman *et al.* 2006; Weigelt *et al.* 2008; Cardinale *et al.* 2013; Wang & Loreau 2016; Wang *et al.* 2017), which suggests that our conclusion that biodiversity decreases spatial variability is robust.’

Question-5:

Finally, I think that the conclusions of this publication are pretty vague and the perspectives inexistent. Do not get me wrong, I believe that the last paragraph is well written and interesting. But, the two first sentences are generic, and the implications for the audience relatively limited. I suggest to summarize the results and then to extend their conclusions to a broader public in order to increase the impact of this publication in Nature Communications. What are the implication of their findings

for conservation biology or ecosystem functioning? What is the main message (beyond the results) that the readership needs to remember?

Response: We added a few words in the last paragraph of the discussion to illustrate the implications of our study: ‘Our findings have important implications for biodiversity conservation. They suggest that biodiversity conservation will be more beneficial to mean ecosystem functioning in low-productivity areas, while it will be more beneficial to the stabilization of ecosystem functioning in high-productivity areas. They also suggest that the effects of biodiversity on ecosystem functioning can only be understood comprehensively on large, or even global, scales. Integrating biodiversity–productivity and biodiversity–stability relationships into a single unified picture at the global extent provides a system-level understanding of ecological processes that is likely to transform ecology into a more systematic science.’

Question-6:

Overall, let me reiterate that the authors satisfied most of my comments and considerably improved the clarity of their manuscript. I believe that this publication is outstanding in its new version and if the two other reviewers agree with the revisions, I would support this work to be published in Nature Communications, which should be of interest to a wide audience.

N. Fanin

Response: Many thanks to these helpful comments.

Reviewer #4 (Remarks to the Author) (Editor's note - Reviewer recruited to comment on behalf of Reviewer #3 regarding the Bayesian approaches):

Question-7:

I enjoyed reviewing this paper and felt the improvements made by the authors in this revision added a lot of clarification. My comments pertain to the details in the Bayesian model and while they are somewhat extensive, I hope the authors find them helpful in terms of improving the manuscript.

The authors posit that biodiversity stabilizes plant productivity across space due to the positive effects of species richness on productivity in harsh environments giving way to constant and finally negative effects as stress decreases (Fig 1 and lines 50-70). In their Bayesian model, these changing biodiversity effects across stress gradients (represented by average grid productivity) are nicely captured using random coefficients where slope is a function of grid productivity (a proxy for the stress all plots within a grid are assumed to be subject to). If spatial stability is in fact occurring due to the mechanism the authors propose, then the reduced variation in productivity seen in species rich plots is due mainly to this convergence of predicted grid productivity as biodiversity increases (this interpretation is taken directly from Fig 1 and lines 50-70).

Based on their Bayesian model, the authors use two key parameter estimates to support the conceptual model visualized in Fig 1. The first is that the γ_1 is negative resulting in the $\alpha_{i,j}$ slope parameters become increasingly negative as grid productivity increases and the second is that within grid plot-level variation decreases as biodiversity increases. While this probably results in convergence resulting in spatial stability across a biodiversity gradient, it is hard

to know from the way the authors have chosen to present their results just how much global variation in plot-level productivity varies with respect to biodiversity, as this depends heavily on the combination of intercept (α_{0j}) and slope (α_{1j}), and range of species richness values for each of the grid productivity gradients. To address this, I suggest the authors use the output from their Bayesian model to do two things: 1) recreate Fig. 1a using the grid-level slope and intercepts posterior means so we can see visually whether your statistical model actually supports the convergence visualized in Fig. 1a and 2) recreate Fig. 1b by predicting plant productivity for equal number of plots from each grid along this biodiversity gradient, computing the spatial stability amongst these plots as a derived quantity in your Bayesian model, and then plotting stability as a function of biodiversity.

Response: We thank the reviewer for these excellent suggestions. We have recreated Fig. 1a and 1b from our fitted Bayesian model according to the reviewer's suggestions. The results (see Fig. 4) are consistent with our hypotheses presented in Fig. 1.

I have the following comments regarding the Bayesian model presented in the methods section:

Question-8:

1) It is standard practice to perform posterior predictive checks (see Gelman and Hill, 2007 or Hobbs and Hooten, 2015) to assess model fit. If the authors haven't already done these then they should, especially since their model is being used mainly to variation in productivity. These checks are easy to do computationally and asking the authors to report these are not onerous.

Response: We have added posterior predictive checks for our models. We provided three posterior predictive p values for assessing the difference between observed and predicted biomass (p values based on the difference in mean and CV of biomass of all plots, and p value based on the difference in biomass of each plot). All these p values are close to 0.5 and indicate good model fits. We also calculated the posterior predictive intervals for the relationship between grid-level biomass and biodiversity effect (Fig. 5) and the relationship between species richness and spatial variability of productivity (Fig. 6).

Question-9:

2) In addition to the code, the authors should provide summaries of the model output in a supplement.

Response: We have supplemented the summary statistics for core parameters of the models as Supplementary Table 1.

Question-10:

3) The R^2 calculation (line 360) captures the amount of variation in logged SD or CV explained by species richness but is not a good metric of overall model fit. Again, I would do posterior predictive checks for this. Also, equation 1c is incorrect as $\alpha_{0,j}$ is modeled hierarchically in your R code. I believe this epsilon should be subscripted with a k and is actually a derived quantity computed as $\log(\sigma_k) - (\beta_0 + \beta_1 * \text{richness}_k)$.

Response: We have deleted R^2 and calculated the posterior predictive intervals as stated above. We have added daylight hours, precipitation and temperature of each grid as predictors of grid-level biomass (α_{0j}). The updated equation related to α_{0j} is: $\alpha_{0j} \sim \text{Normal}(\gamma_{00} + \gamma_{01} * \text{daylight}_j + \gamma_{02} * \text{temperature}_j +$

$\text{gamma0}_3 * \text{precipitation}_j, \text{alpha}_0_{\text{sigma}}).$

Question-11:

4) Equations 1a-1d do not match the R code as they are missing the random intercept model. Also $\text{alpha}_{1,j}$ does = normal it is \sim normal. I really would present the model in these equations using a joint distribution, as described in Hobbs and Hooten 2015, but I understand if you leave it presented in this fashion (after correcting).

Response: We have corrected the equation as the reviewer suggested. We modeled alpha_{0j} and alpha_{1j} as independent distributions instead of a joint distribution because we already modeled the correlation between these two parameters in equation 1d (where grid-level average productivity is a predictor of alpha_{1j} — note that alpha_{0j} represents grid-level average productivity.).

Question-12:

5) Productivity is strictly positive and given that you have values close to zero I would suggest using the lognormal or gamma distribution instead of the normal to model productivity instead. I don't think this would require you to change the distributions for the other parameters. Also, the biodiversity effects being modeled here ($\text{alpha}_{1,j}$) are all additive making them hard to compare across grid-level productivity gradients. If you used the lognormal or gamma distribution, it would be easy to use the log-link function to model these biodiversity effects as multiplicative, which are much easier to compare across productivity/stress gradients.

Response: We agree with the reviewer that using a lognormal or gamma distribution might be a good alternative approach. However, our models using normal distribution produced quite good model fits based on the posterior predictive checks. Using normal distribution is also more straightforward for testing our hypotheses. We can easily calculate the SD or CV of untransformed biomass (instead of the SD or CV of log-transformed biomass) as measures of spatial stability in the models, which are mostly used in the recent literature. We are interested in the effect of adding one species on the absolute change of productivity (Fig. 1a). Our hypothesis is about how this additive diversity effect will drive the spatial stability of biomass. It is important that we use biomass instead of other transformed measures of biomass as our response variable. Therefore, we kept the current model based on normal distribution and additive diversity effect.

Question-13:

6) I am not clear why your model does not include other covariates from the SEM known to affect plot level productivity, such as daylight hours, temperature, or precipitation, as covariates in Equation 1d?

Response: We have now included daylight hours, precipitation and daylight hours as predictors of grid-level productivity in our Bayesian models.

Question-14:

7) I agree with your choice of plot- and grid-level for this analysis (response to Q19/20). However, there is likely non-independence amongst the grids due to site effects and you should consider adding a random site effect to account for this.

Response: We have now added a random site effect in our Bayesian models.

Question-15:

8) In Fig. 1a there is a clear correlation between the slope and intercepts (more productive grids have flatter or more negative slopes but larger intercept values), but in your random coefficients model you do not model this correlation at all. Given that such a negative correlation btw slope and intercept is needed to get these biodiversity-productivity lines to converge as in Fig. 1a, modeling it would be helpful.

Response: We agree with the reviewer that there is correlation between the slope (α_{1j}) and intercept (α_{0j}). We have modeled this correlation in equation 1d, where we have grid-level productivity as a predictor for the slope. Note that α_{0j} represents the grid-level average productivity.

Question-16:

9) Lines 353 and 357: What do you mean by first and second hypothesis? Searching earlier in the text, I can't find a reference to two separate hypotheses being tested by the Bayesian model.

Response: The two hypotheses were presented in Introduction. We have revised the text to make the hypotheses clearer.

Question-17:

10) How is the Bayesian model described in lines 125-132 different from the Bayesian model described in lines 115-118, Figure 4, and extensively in the methods? If the Bayesian model present in Fig. 5b,d is the same as the one previously described earlier lines 115-118, then I would omit the simple linear regressions. If it is not, then I would omit the new Bayesian analysis.

Response: Lines 125-132 (i.e., Fig. 5b,d) correspond to equation 1b and present the relationship between richness and spatial stability (corresponding to the hypothesis presented in Fig. 1b). Lines 115-118 (i.e., Fig. 4) correspond to equation 1d and present the relationship between grid-level average productivity and richness effect (corresponding to the hypothesis presented in Fig. 1a). Thus, the Bayesian model presented in Fig. 5b,d is different from the one described in lines 115-118 (i.e., Fig. 4).

Regarding questions raised by the Reviewer #3

Q17: Adequately addressed by the reviewers.

Question-18:

Q18: I had the same concern as Reviewer #3 when reading this manuscript (especially after seeing the proportion of variation explained in Fig. 2). The authors responded that because there was substantial variation in the grid-specific slope parameters and this “made richness a strong predictor for spatial stability of biomass at the global scale, which is consistent with our hypotheses.” This might be true (see my main comment above) but the authors should compute Bayesian R^2 to see the proportion of plot-level productivity explained by grid-level richness effects to really address the reviewer's concern. Again, this would be straightforward to add to the model code.

Response: We agree with Reviewer #3 that species richness is not important for average

productivity at the plot level, which is indicated by the low R^2 (< 10% by climate variables and richness; Fig. 2) in the SEM. But the point in our previous response letter is that species richness is important for variation of productivity at each richness level, which is indicated by the high R^2 (67% - 83%) in Fig. 6 of the previous version.

Question-19:

Q19/Q20: I agree with the authors choice of grid- and plot level for this analysis. However, I share the concern of the previous reviewer, that since grids are nested within sites are likely non-independent and the authors should also add a site-level random effect.

***Response:* We have added a random site effect in our Bayesian models.**

Q21/Q22: Adequately addressed by the reviewers.

REVIEWERS' COMMENTS:

Reviewer #4 (Remarks to the Author):

I have the following comments regarding the Bayesian model presented in the methods section:

1) It is standard practice to perform posterior predictive checks (see Gelman and Hill, 2007 or Hobbs and Hooten, 2015) to assess model fit. If the authors haven't already done these then they should, especially since their model is being used mainly to variation in productivity. These checks are easy to do computationally and asking the authors to report these are not onerous.

Authors response: We have added posterior predictive checks for our models. We provided three posterior predictive p values for assessing the difference between observed and predicted biomass (p values based on the difference in mean and CV of biomass of all plots, and p value based on the difference in biomass of each plot). All these p values are close to 0.5 and indicate good model fits. We also calculated the posterior predictive intervals for the relationship between grid-level biomass and biodiversity effect (Fig. 5) and the relationship between species richness and spatial variability of productivity (Fig. 6).

Reviewer response: Adequately addressed by the authors.

2) In addition to the code, the authors should provide summaries of the model output in a supplement.

Author Response: We have supplemented the summary statistics for core parameters of the models as Supplementary Table 1.

Reviewer response: Adequately addressed by the authors.

3) The R^2 calculation (line 360) captures the amount of variation in logged SD or CV explained by species richness but is not a good metric of overall model fit. Again, I would do posterior predictive checks for this. Also, equation 1c is incorrect as $\alpha_{0,j}$ is modeled hierarchically in your R code. I believe this epsilon should be subscripted with a k and is actually a derived quantity computed as $\log(\sigma_k) - (\beta_0 + \beta_1 * \text{richness}_k)$.

Author Response: We have deleted R^2 and calculated the posterior predictive intervals as stated above. We have added daylight hours, precipitation and temperature of each grid as predictors of grid-level biomass (α_{0j}). The updated equation related to α_{0j} is: $\alpha_{0j} \sim \text{Normal}(\gamma_{0_0} + \gamma_{0_1} * \text{daylight}_j + \gamma_{0_2} * \text{temperature}_j + \gamma_{0_3} * \text{precipitation}_j, \alpha_{0_sigma})$.

Reviewer response: Adequately addressed by the authors.

4) Equations 1a-1d do not match the R code as they are missing the random intercept model. Also $\alpha_{1,j}$ does = normal it is \sim normal. I really would present the model in these equations using a joint distribution, as described in Hobbs and Hooten 2015, but I understand if you leave it presented in this fashion (after correcting).

Author Response: We have corrected the equation as the reviewer suggested. We modeled α_{0j} and α_{1j} as independent distributions instead of a joint distribution because we already modeled the correlation between these two parameters in equation 1d (where grid-level average productivity is a predictor of α_{1j} — note that α_{0j} represents grid-level average productivity.).

Reviewer Response: I am not sure equation 1d is modeling the correlation between the grid-level intercept and slopes. To do that you would model them as MVN with a covariance parameter on the off-diagonal element on the variance-covariance matrix (see Gelman and Hill, ch 13 or any other statistical text on hierarchical modeling). This would capture the negative correlation btw

intercept and slope you portray in Figure 1a and you actually see in Fig 4a, where the relationship btw richness and productivity (at the plot level) varies as a function of grid productivity such that low productive grids have steeper slopes and smaller intercepts. Nonetheless, I see your reasoning here - by including grid productivity in 1d (such that the grid productivity_j is essentially equivalent to alpha_{0,j}), this effectively models the slope as a function of the intercept. I think the way you have it now is ok, especially since you are not predicting new grids and plots from your model to make inferences, but instead are relying on the fitted grids and plots to test your hypotheses in Fig 4a and b.

5) Productivity is strictly positive and given that you have values close to zero I would suggest using the lognormal or gamma distribution instead of the normal to model productivity instead. I don't think this would require you to change the distributions for the other parameters. Also, the biodiversity effects being modeled here (alpha_{1,j}) are all additive making them hard to compare across grid-level productivity gradients. If you used the lognormal or gamma distribution, it would be easy to use the log-link function to model these biodiversity effects as multiplicative, which are much easier to compare across productivity/stress gradients.

Author response: We agree with the reviewer that using a lognormal or gamma distribution might be a good alternative approach. However, our models using normal distribution produced quite good model fits based on the posterior predictive checks. Using normal distribution is also more straightforward for testing our hypotheses. We can easily calculate the SD or CV of untransformed biomass (instead of the SD or CV of log-transformed biomass) as measures of spatial stability in the models, which are mostly used in the recent literature. We are interested in the effect of adding one species on the absolute change of productivity (Fig. 1a). Our hypothesis is about how this additive diversity effect will drive the spatial stability of biomass. It is important that we use biomass instead of other transformed measures of biomass as our response variable. Therefore, we kept the current model based on normal distribution and additive diversity effect.

Reviewer response: I agree that the posterior predictive checks suggest that the normal model fits the data well and I would accept that the authors use of normal vs. positive only distributions is ok here. It is worth noting that you do not need to log transform biomass to use the lognormal or gamma distributions (which address your concern that working with log-transformed biomass is unwieldy), and even if you did you could back transform it as derived quantity before computing SD or CV. Also, I still think that interpretation of the coefficients would be better suited as multiplicative than additive. For instance, a 1 unit change in productivity per species added represents a much more drastic change in low than high productive environments (I am looking at the range in productivity on Fig 4a), whereas a multiplicative change essentially "rescales" these effects to be consistent across all productivities. I am not raising these points to be argumentative or draw a line in the sand here on this analysis that you have included, but mainly just to point out how I think, despite the reassurance of the posterior predictive checks and your desire to avoid logs, your message would be better communicated with a lognormal or gamma model.

6) I am not clear why your model does not include other covariates from the SEM known to affect plot level productivity, such as daylight hours, temperature, or precipitation, as covariates in Equation 1d?

Author Response: We have now included daylight hours, precipitation and daylight hours as predictors of grid-level productivity in our Bayesian models.

Reviewer Response: If I am understanding you correctly, in equation 1c you predict grid-level productivity as a function of daylight hours precipitation and temperature, so grids that have more favorable combinations of these covariates are likely to have higher grid productivity and hence a_{0,j} for those plots contained within that grid are likely to have higher productivity at 0 species richness (this is what the intercept is representing here unless you standardized richness, which is unclear). Instead, I suggest including light, temperature, and precipitation as covariates in both equations 1c and 1d as covariates. The advantage here is that after fitting this model you (or anyone else using your model output) could predict species richness/productivity relationships and spatial stability for any combination of these climatic variables, which would be really interesting. Alternatively, you could just use grid productivity alone as a single covariate in both 1c and 1d, but

I think this limits the predictive ability of this model compared to using light, precipitation, and temperature.

7) I agree with your choice of plot- and grid-level for this analysis (response to Q19/20). However, there is likely non-independence amongst the grids due to site effects and you should consider adding a random site effect to account for this.

Author Response: We have now added a random site effect in our Bayesian models.

Reviewer response: Adequately addressed by the authors.

8) In Fig. 1a there is a clear correlation between the slope and intercepts (more productive grids have flatter or more negative slopes but larger intercept values), but in your random coefficients model you do not model this correlation at all. Given that such a negative correlation btw slope and intercept is needed to get these biodiversity-productivity lines to converge as in Fig. 1a, modeling it would be helpful.

Author response: We agree with the reviewer that there is correlation between the slope (α_{1j}) and intercept (α_{0j}). We have modeled this correlation in equation 1d, where we have grid-level productivity as a predictor for the slope. Note that α_{0j} represents the grid-level average productivity.

Reviewer response: Adequately addressed by the authors - see response to #4 above.

9) Lines 353 and 357: What do you mean by first and second hypothesis? Searching earlier in the text, I can't find a reference to two separate hypotheses being tested by the Bayesian model.

Author Response: The two hypotheses were presented in Introduction. We have revised the text to make the hypotheses clearer.

Reviewer response: Adequately addressed by the authors.

10) How is the Bayesian model described in lines 125-132 different from the Bayesian model described in lines 115-118, Figure 4, and extensively in the methods? If the Bayesian model present in Fig. 5b,d is the same as the one previously described earlier lines 115-118, then I would omit the simple linear regressions. If it is not, then I would omit the new Bayesian analysis.

Author Response: Lines 125-132 (i.e., Fig. 5b,d) correspond to equation 1b and present the relationship between richness and spatial stability (corresponding to the hypothesis presented in Fig. 1b). Lines 115-118 (i.e., Fig. 4) correspond to equation 1d and present the relationship between grid-level average productivity and richness effect (corresponding to the hypothesis presented in Fig. 1a). Thus, the Bayesian model presented in Fig. 5b,d is different from the one described in lines 115-118 (i.e., Fig. 4).

Reviewer response: Adequately addressed by the authors.

Regarding questions raised by the Reviewer #3

Q17: Adequately addressed by the authors.

Q18: I had the same concern as Reviewer #3 when reading this manuscript (especially after seeing the proportion of variation explained in Fig. 2). The authors responded that because there was substantial variation in the grid-specific slope parameters and this "made richness a strong predictor for spatial stability of biomass at the global scale, which is consistent with our hypotheses." This might be true (see my main comment above) but the authors should compute Bayesian R^2 to see the proportion of plot-level productivity explained by grid-level richness effects to really address the reviewer's concern. Again, this would be straightforward to add to the model code.

Author Response: We agree with Reviewer #3 that species richness is not important for average productivity at the plot level, which is indicated by the low R^2 ($< 10\%$ by climate variables and richness; Fig. 2) in the SEM. But the point in our previous response letter is that species richness is important for variation of productivity at each richness level, which is indicated by the high R^2 (67% - 83%) in Fig. 6 of the previous version.

Q19/Q20: I agree with the authors choice of grid- and plot level for this analysis. However, I share the concern of the previous reviewer, that since grids are nested within sites are likely non-independent and the authors should also add a site-level random effect.

Author response: We have added a random site effect in our Bayesian models.

Reviewer response: Adequately addressed by the authors.

Q21/Q22: Adequately addressed by the authors.

REVIEWERS' COMMENTS:

Reviewer #4 (Remarks to the Author):

I have the following comments regarding the Bayesian model presented in the methods section:

1) It is standard practice to perform posterior predictive checks (see Gelman and Hill, 2007 or Hobbs and Hooten, 2015) to assess model fit. If the authors haven't already done these then they should, especially since their model is being used mainly to variation in productivity. These checks are easy to do computationally and asking the authors to report these are not onerous.

Authors response: We have added posterior predictive checks for our models. We provided three posterior predictive p values for assessing the difference between observed and predicted biomass (p values based on the difference in mean and CV of biomass of all plots, and p value based on the difference in biomass of each plot). All these p values are close to 0.5 and indicate good model fits. We also calculated the posterior predictive intervals for the relationship between grid-level biomass and biodiversity effect (Fig. 5) and the relationship between species richness and spatial variability of productivity (Fig. 6).

Reviewer response: Adequately addressed by the authors.

2) In addition to the code, the authors should provide summaries of the model output in a supplement.

Author Response: We have supplemented the summary statistics for core parameters of the models as Supplementary Table 1.

Reviewer response: Adequately addressed by the authors.

3) The R^2 calculation (line 360) captures the amount of variation in logged SD or CV explained by species richness but is not a good metric of overall model fit. Again, I would do posterior predictive checks for this. Also, equation 1c is incorrect as $\alpha_{0,j}$ is modeled hierarchically in your R code. I believe this epsilon should be subscripted with a k and is actually a derived quantity computed as $\log(\sigma_k) - (\beta_0 + \beta_1 * \text{richness}_k)$.

Author Response: We have deleted R^2 and calculated the posterior predictive intervals as stated above. We have added daylight hours, precipitation and temperature of each grid as predictors of grid-level biomass (α_{0j}). The updated equation

related to α_{0j} is: $\alpha_{0j} \sim \text{Normal}(\gamma_{0_0} + \gamma_{0_1} * \text{daylight}_j + \gamma_{0_2} * \text{temperature}_j + \gamma_{0_3} * \text{precipitation}_j, \alpha_{0_sigma})$.

Reviewer response: Adequately addressed by the authors.

4) Equations 1a-1d do not match the R code as they are missing the random intercept model. Also $\alpha_{1,j}$ does = normal it is \sim normal. I really would present the model in these equations using a joint distribution, as described in Hobbs and Hooten 2015, but I understand if you leave it presented in this fashion (after correcting).

Author Response: We have corrected the equation as the reviewer suggested. We modeled α_{0j} and α_{1j} as independent distributions instead of a joint distribution because we already modeled the correlation between these two parameters in equation 1d (where grid-level average productivity is a predictor of α_{1j} — note that α_{0j} represents grid-level average productivity.).

Reviewer Response: I am not sure equation 1d is modeling the correlation between the grid-level intercept and slopes. To do that you would model them as MVN with a covariance parameter on the off-diagonal element on the variance-covariance matrix (see Gelman and Hill, ch 13 or any other statistical text on hierarchical modeling). This would capture the negative correlation btw intercept and slope you portray in Figure 1a and you actually see in Fig 4a, where the relationship btw richness and productivity (at the plot level) varies as a function of grid productivity such that low productive grids have steeper slopes and smaller intercepts. Nonetheless, I see your reasoning here - by including grid productivity in 1d (such that the grid productivity_j is essentially equivalent to $\alpha_{0,j}$), this effectively models the slope as a function of the intercept. I think the way you have it now is ok, especially since you are not predicting new grids and plots from your model to make inferences, but instead are relying on the fitted grids and plots to test your hypotheses in Fig 4a and b.

Authors Response: We agree that both MVN and equation 1d (now it is equation 4 in the new version) of our model can model the correlation between the grid-level intercept and slope. We keep equation 1d because it is easy to model and interpret. Having grid productivity as a predictor for α_{1j} , it is straightforward to test the hypothesis that how biodiversity effects vary across grid productivity gradient (Fig. 5).

5) Productivity is strictly positive and given that you have values close to zero I would suggest using the lognormal or gamma distribution instead of the normal to model productivity instead. I don't think this would require you to change the distributions for the other parameters. Also, the biodiversity effects being modeled here ($\alpha_{1,j}$) are all additive making them hard to compare across grid-level productivity gradients. If you used the lognormal or gamma distribution, it would be easy to use the log-link

function to model these biodiversity effects as multiplicative, which are much easier to compare across productivity/stress gradients.

Author response: We agree with the reviewer that using a lognormal or gamma distribution might be a good alternative approach. However, our models using normal distribution produced quite good model fits based on the posterior predictive checks. Using normal distribution is also more straightforward for testing our hypotheses. We can easily calculate the SD or CV of un-transformed biomass (instead of the SD or CV of log-transformed biomass) as measures of spatial stability in the models, which are mostly used in the recent literature. We are interested in the effect of adding one species on the absolute change of productivity (Fig. 1a). Our hypothesis is about how this additive diversity effect will drive the spatial stability of biomass. It is important that we use biomass instead of other transformed measures of biomass as our response variable. Therefore, we kept the current model based on normal distribution and additive diversity effect.

Reviewer response: I agree that the posterior predictive checks suggest that the normal model fits the data well and I would accept that the authors use of normal vs. positive only distributions is ok here. It is worth noting that you do not need to log transform biomass to use the lognormal or gamma distributions (which address your concern that working with log-transformed biomass is unwieldy), and even if you did you could back transform it as derived quantity before computing SD or CV. Also, I still think that interpretation of the coefficients would be better suited as multiplicative than additive. For instance, a 1 unit change in productivity per species added represents a much more drastic change in low than high productive environments (I am looking at the range in productivity on Fig 4a), whereas a multiplicative change essentially "rescales" these effects to be consistent across all productivities. I am not raising these points to be argumentative or draw a line in the sand here on this analysis that you have included, but mainly just to point out how I think, despite the reassurance of the posterior predictive checks and your desire to avoid logs, your message would be better communicated with a lognormal or gamma model.

Authors Response: We thank the reviewer for the detailed explanation of the alternative analysis. We used the normal model and additive biodiversity effect because it produces a very good model fit and straightforward to test our hypothesis on absolute (not relative) change in biomass (Figs. 1 and 4).

6) I am not clear why your model does not include other covariates from the SEM known to affect plot level productivity, such as daylight hours, temperature, or precipitation, as covariates in Equation 1d?

Author Response: We have now included daylight hours, precipitation and daylight hours as predictors of grid-level productivity in our Bayesian models.

Reviewer Response: If I am understanding you correctly, in equation 1c you predict grid-level productivity as a function of daylight hours precipitation and temperature, so grids that have more favorable combinations of these covariates are likely to have higher grid productivity and hence $\alpha_{0,j}$ for those plots contained within that grid are likely to have higher productivity at 0 species richness (this is what the intercept is representing here unless you standardized richness, which is unclear). Instead, I suggest including light, temperature, and precipitation as covariates in both equations 1c and 1d as covariates. The advantage here is that after fitting this model you (or anyone else using your model output) could predict species richness/productivity relationships and spatial stability for any combination of these climatic variables, which would be really interesting. Alternatively, you could just use grid productivity alone as a single covariate in both 1c and 1d, but I think this limits the predictive ability of this model compared to using light, precipitation, and temperature.

***Authors Response:* We thank the reviewer for the detailed explanation for the alternative analysis. We used grid productivity as a single predictor in equation 1d (now it is equation 4 in the new version) because we want to test the hypothesis that how biodiversity effects vary across grid productivity gradient (Fig. 5). This is more important than for predicting biodiversity effects under new sites with known climatic conditions in our case.**

7) I agree with your choice of plot- and grid-level for this analysis (response to Q19/20). However, there is likely non-independence amongst the grids due to site effects and you should consider adding a random site effect to account for this.

Author Response: We have now added a random site effect in our Bayesian models.

Reviewer response: Adequately addressed by the authors.

8) In Fig. 1a there is a clear correlation between the slope and intercepts (more productive grids have flatter or more negative slopes but larger intercept values), but in your random coefficients model you do not model this correlation at all. Given that such a negative correlation btw slope and intercept is needed to get these biodiversity-productivity lines to converge as in Fig. 1a, modeling it would be helpful.

Author response: We agree with the reviewer that there is correlation between the slope (α_{1j}) and intercept (α_{0j}). We have modeled this correlation in equation 1d, where we have grid-level productivity as a predictor for the slope. Note that α_{0j} represents the grid-level average productivity.

Reviewer response: Adequately addressed by the authors - see response to #4 above.

9) Lines 353 and 357: What do you mean by first and second hypothesis? Searching earlier in the text, I can't find a reference to two separate hypotheses being tested by the Bayesian model.

Author Response: The two hypotheses were presented in Introduction. We have revised the text to make the hypotheses clearer.

Reviewer response: Adequately addressed by the authors.

10) How is the Bayesian model described in lines 125-132 different from the Bayesian model described in lines 115-118, Figure 4, and extensively in the methods? If the Bayesian model present in Fig. 5b,d is the same as the one previously described earlier lines 115-118, then I would omit the simple linear regressions. If it is not, then I would omit the new Bayesian analysis.

Author Response: Lines 125-132 (i.e., Fig. 5b,d) correspond to equation 1b and present the relationship between richness and spatial stability (corresponding to the hypothesis presented in Fig. 1b). Lines 115-118 (i.e., Fig. 4) correspond to equation 1d and present the relationship between grid-level average productivity and richness effect (corresponding to the hypothesis presented in Fig. 1a). Thus, the Bayesian model presented in Fig. 5b,d is different from the one described in lines 115-118 (i.e., Fig. 4).

Reviewer response: Adequately addressed by the authors.

Regarding questions raised by the Reviewer #3

Q17: Adequately addressed by the authors.

Q18: I had the same concern as Reviewer #3 when reading this manuscript (especially after seeing the proportion of variation explained in Fig. 2). The authors responded that because there was substantial variation in the grid-specific slope parameters and this “made richness a strong predictor for spatial stability of biomass at the global scale, which is consistent with our hypotheses.” This might be true (see my main comment above) but the authors should compute Bayesian R^2 to see the proportion of plot-level productivity explained by grid-level richness effects to really address the reviewer's concern. Again, this would be straightforward to add to the model code.

Author Response: We agree with Reviewer #3 that species richness is not important for average productivity at the plot level, which is indicated by the low R^2 (< 10% by climate variables and richness; Fig. 2) in the SEM. But the point in our previous response letter is that species richness is important for variation of productivity at each richness level, which is indicated by the high R^2 (67% - 83%) in Fig. 6 of the previous version.

Q19/Q20: I agree with the authors choice of grid- and plot level for this analysis. However, I share the concern of the previous reviewer, that since grids are nested within sites are likely non-independent and the authors should also add a site-level random effect.

Author response: We have added a random site effect in our Bayesian models.

Reviewer response: Adequately addressed by the authors.

Q21/Q22: Adequately addressed by the authors.